# Lightweight-Mark: Rethinking Deep Learning-Based Watermarking

Yupeng Qiu [1]    Han Fang [1 *]    Ee-Chien Chang [1]

## Abstract

Deep learning-based watermarking models play a crucial role in copyright protection across various applications. However, many high-performance models are limited in practical deployment due to their large number of parameters. Meanwhile, the robustness and invisibility performance of existing lightweight models are unsatisfactory. This presents a pressing need for a watermarking model that combines lightweight capacity with satisfactory performance. Our research identifies a key reason that limits the performance of existing watermarking frameworks: a mismatch between commonly used decoding losses (e.g., mean squared error and binary cross-entropy loss) and the actual decoding goal, leading to parameter redundancy. We propose two innovative solutions: (1) Decoding-oriented surrogate loss (DO), which redesigns the loss function to mitigate the influence of decoding-irrelevant optimization directions; and (2) Detachable projection head (PH), which incorporates a detachable redundant module during training to handle these irrelevant directions and is discarded during inference. Additionally, we propose a novel watermarking framework comprising five submodules, allowing for independent parameter reduction in each component. Our proposed model achieves better efficiency, invisibility, and robustness while utilizing only 2.2% of the parameters compared to the state-of-the-art frameworks. By improving efficiency while maintaining robust copyright protection, our model is well suited for practical applications in resource-constrained environments. The DO and PH methods are designed to be plug-and-play, facilitating seamless integration into future lightweight models.

*Corresponding Author. [1]National University of Singapore. Correspondence to: Han Fang <fanghan@nus.edu.sg>.

*Proceedings of the $42^{nd}$ International Conference on Machine Learning*, Vancouver, Canada. PMLR 267, 2025. Copyright 2025 by the author(s).

## 1. Introduction

Digital watermarking is crucial for protecting the copyright of various digital assets, including images, videos, and 3D content. Typically, digital watermarking involves two main tasks: information hiding and extraction. In this process, digital media owners conceal secret information within the digital media, which can later be extracted to authenticate copyright ownership. In this process, two crucial properties need to be considered: **1) invisibility**, which requires that the visual quality of the digital media does not significantly degrade after embedding secret information, and **2) robustness**, which ensures that even when watermarked digital media encounter various distortions, the embedded information can be accurately extracted (Wan et al., 2022; Singh, 2023).

In recent years, with the development of deep learning, many deep learning-based watermarking models have emerged, which utilize complex architectures to improve performance (Liu et al., 2019; Zhang et al., 2021; Jia et al., 2021; Ma et al., 2022; Fang et al., 2023). However, these high-performance models often come with a large number of parameters and huge computational demands, limiting their practical deployment. As a result, fields such as diffusion models (Fernandez et al., 2023; Zhao et al., 2023b) and Neural Radiance Fields (NeRF) (Luo et al., 2023; Jang et al., 2024) have chosen to adopt lightweight watermarking models like HiDDeN (Zhu et al., 2018) for copyright protection, as their backbone models are already large and computationally intensive. Although lightweight models are easy to deploy, they often sacrifice robustness. This limitation impedes effective and efficient copyright protection in numerous fields, especially where computational resources are limited, highlighting an urgent need for watermarking models that combine lightweight capacity with satisfactory performance.

This paper aims to design and train a lightweight watermarking model with state-of-the-art performance. To understand the potential reasons limiting the performance of lightweight watermarking models, we investigate the decoding process. For the decoding task, accuracy serves as a prevalent evaluation metric. However, the straightforward empirical risk minimization (ERM) formulation for accuracy includes minimizing the 0-1 loss, which is computationally intractable. Consequently, researchers resort

to differentiable losses (e.g., mean squared error or binary cross-entropy loss) as tractable surrogate losses, transforming the decoding task into a reconstruction task (Zhu et al., 2018; Liu et al., 2019; Zhang et al., 2021; Jia et al., 2021; Ma et al., 2022; Fang et al., 2023). The feasibility of optimization comes at the expense of model efficiency. Our observation reveals a mismatch between the actual decoding goal and the optimization objectives of the commonly used decoding losses. By dissecting the decoding losses into deflation, inflation, and regularization losses, we identify that decoding accuracy depends primarily on the deflation loss. However, inflation and regularization losses, while stabilizing training, inevitably consume model parameters, negatively affecting the efficiency of lightweight models. To address this issue, we propose two methods. For the first method, we append an additional projection module to the lightweight model during training. This module manages decoding-irrelevant optimization directions and is discarded during inference. In the other approach, we propose a new surrogate loss to mitigate the negative impact of decoding-irrelevant optimization directions while ensuring stable training.

Moreover, previous works typically treat encoders and decoders as basic design units and often lack detailed categorization of internal functions. This makes it challenging to conduct fine-grained ablation studies. To address this problem, we propose to subdivide encoders and decoders into smaller functional units and, for the first time, summarize a new deep watermarking framework consisting of five modules. We construct a lightweight watermarking model within this new framework using only transposed convolution and convolution layers. Experimental results reveal that certain modules are crucial when facing specific distortions while contributing minimally to other distortions. This insight allows us to selectively remove non-essential modules, further compressing the model with minimal performance degradation.

The main contributions of this paper are summarized as follows: **1)** We are the first to identify the mismatch between the optimization objectives of the commonly used decoding losses (e.g., mean squared error and binary cross-entropy loss) and the actual decoding goal. Ablation studies confirm the presence and impact of this mismatch. **2)** We propose the detachable projection head (PH) and decoding-oriented surrogate loss (DO) to mitigate the negative impact of irrelevant optimization directions, enabling lightweight models to achieve state-of-the-art performance. **3)** We introduce a fine-grained deep watermarking framework with five modules. Our experiments analyze the roles of different modules under various distortions, enabling further model compression with minimal performance loss. **4)** Our lightweight model will see broader application in other fields, especially where computational resources are constrained. **5)** Extensive experiments demonstrate the superiority of our approach over existing models in terms of invisibility, robustness, and efficiency.

## 2. Related Works and Preliminaries

### 2.1. Related Works

**Image Watermarking** Digital watermarking (Van Schyndel et al., 1994) is widely used to protect copyrights and trace the origins of unauthorized copies. Invisibility and robustness are crucial properties of digital watermarks. In pursuit of balancing invisibility and robustness, traditional digital watermarking algorithms often embed watermarks in mid-frequency domain coefficients. The commonly used domains include the discrete cosine transform (DCT) domain (Ahmidi & Safabakhsh, 2004), the discrete wavelet transform (DWT) domain (Daren et al., 2001), and the discrete Fourier transform (DFT) domain (Hamidi et al., 2018). With the advancement of deep learning techniques, deep learning-based watermarking models have received increasing attention for achieving a better trade-off between invisibility and robustness. The first deep learning-based watermarking model was introduced by Kandi et al. (2017), demonstrating the feasibility of using autoencoder convolutional neural networks (CNN) for watermarking tasks. Subsequently, Zhu et al. (2018) proposed the Encoder-NoiseLayer-Decoder (END) framework, pioneering end-to-end training and incorporating a noise layer to enhance robustness against various distortions. Jia et al. (2021) introduced the Mini-Batch of Simulated and Real Jpeg compression (MBRS) training method and advanced Squeeze-and-Excitation (SE) blocks (Hu et al., 2018) to improve robustness against JPEG compression. Ma et al. (2022) developed the Combining Invertible and Non-invertible Mechanisms (CIN), leveraging invertible neural networks for embedding and extraction, significantly improving invisibility and robustness. Fang et al. (2023) proposed the Flow-based Invertible Network (FIN), which explored the use of invertible structures as differentiable simulators for both white-box and black-box distortions, surpassing many state-of-the-art END-based watermarking models. However, in pursuit of better invisibility and robustness, the watermarking models have become bigger, with increased computational complexity.

**Knowledge Distillation** Recently, deep learning has achieved significant success in many fields. These substantial advancements are mainly due to the massive number of model parameters. Although large-scale models exhibit better performance, the massive storage requirements and high computational complexity hinder their further deployment. Therefore, knowledge distillation (KD) has received significant attention, which aims to distill knowledge from a larger model (teacher model) into a smaller model (student model)

to achieve model compression (Hinton et al., 2015; Kim et al., 2018; Mirzadeh et al., 2020; Zhao et al., 2022). Although knowledge distillation has been successfully applied in various applications such as visual recognition, speech recognition, and natural language processing, the theoretical understanding of knowledge distillation is limited (Cho & Hariharan, 2019; Cheng et al., 2020), and the reasons for its success are not fully clear. Moreover, when there is a significant difference in model architecture or size between the student model and the teacher model, this model capacity gap can degrade knowledge transfer (Mirzadeh et al., 2020; Gao et al., 2021). This implies that typically, the student model performs inferior to the teacher model. Therefore, obtaining a lightweight model with state-of-the-art performance through knowledge distillation is challenging.

## 2.2. Preliminaries

**Notations** This paper defines several key notions for clarity and consistency in subsequent discussions. The watermark, denoted as $M \in \{-1, 1\}^L$, and the extracted watermark, denoted as $M_{\mathrm{ex}} \in \mathbb{R}^L$, are both represented as message sequences of length $L$. The cover image $I_{\mathrm{co}}$, watermarked image $I_{\mathrm{w}}$, and noised watermarked image $I_{\mathrm{no}}$ are all RGB images that belong to $\mathbb{R}^{C \times W \times H}$, where $W$ and $H$ denote the width and height, respectively. $\mathcal{M} \subseteq \{-1, 1\}^L$ and $\mathcal{M}_{\mathrm{ex}} \subseteq \mathbb{R}^L$, denoted as the support of $M$ and $M_{\mathrm{ex}}$. $\mathcal{I}_{\mathrm{co}}$, $\mathcal{I}_{\mathrm{w}}$, and $\mathcal{I}_{\mathrm{no}} \subseteq \mathbb{R}^{C \times W \times H}$, denoted as the support of $I_{\mathrm{co}}$, $I_{\mathrm{w}}$, and $I_{\mathrm{no}}$. Denote by $f : (\mathcal{M}, \mathcal{I}_{\mathrm{co}}) \to \mathcal{I}_{\mathrm{w}}$ the information hiding function, which embeds a secret message into a cover image. Denote by $g : \mathcal{I}_{\mathrm{no}} \to \mathcal{M}_{\mathrm{ex}}$ the information extraction function, which extracts a secret message from a noised watermarked image. These two functions are parameterized by deep neural networks. The indicator function $\mathbf{1}\{\text{event}\}$, also known as the 0-1 loss, denotes an indicator function that outputs 1 if an event happens and 0 otherwise. $\mathcal{L}_{\mathrm{visual}} (I_{\mathrm{co}}, I_{\mathrm{w}}) = MSE (I_{\mathrm{co}}, I_{\mathrm{w}})$ is the visual loss used to ensure visual quality, whose goal is to make the watermarked image closely resemble the cover image.

**Robustness (accuracy)** In the information extraction stage, the decoder $g$ obtains the extracted watermark $M_{\mathrm{ex}}$ from a noised watermarked image $I_{\mathrm{no}}$. Researchers usually use accuracy or decoding error to characterize the decoder's robustness. Maximizing accuracy is equivalent to minimizing the decoding error. The decoding error can be defined as follows:

$$Error (g(I_{\mathrm{no}}), M) = \mathbb{E} \left[ \frac{1}{L} \sum_{i=1}^{L} \mathbf{1}\{g_i(I_{\mathrm{no}}) \cdot M_i < 0\} \right] \tag{1}$$

Here $g_i(\cdot)$ and $M_i$ denote the $i^{th}$ bit in $g(\cdot)$ and $M$.

**The Gap between Two Objectives** As a metric for measuring robustness, decoding error is straightforward. However, minimizing Equation (1) using deep learning models is computationally intractable since it cannot be optimized by the gradient descent algorithm. Therefore, researchers minimize a differentiable surrogate loss to address this challenge. Typically, this chosen surrogate loss serves as an upper bound for the decoding error, and Bartlett et al. (2006) ensured that minimizing this differentiable upper bound helps to reduce the decoding error. A commonly used surrogate loss is the mean squared error (MSE) loss, which can be represented as follows. For a more detailed analysis of the binary cross-entropy loss (BCE), please refer to Appendix A.2.

$$MSE (g(I_{\mathrm{no}}), M) = \mathbb{E} \left[ \frac{1}{L} \sum_{i=1}^{L} (g_i(I_{\mathrm{no}}) - M_i)^2 \right] \tag{2}$$

However, minimizing the MSE loss does not perfectly match the objective of reducing the decoding error. To better understand the impact of this gap on model performance, we dissect the MSE loss into seven components:

$$MSE (g(I_{\mathrm{no}}), M) = \frac{1}{L} \mathbb{E} \Bigg[ \underbrace{\sum_{i=1}^{L_{\mathrm{R}}} g_i^2(I_{\mathrm{no}})}_{\mathcal{L}_{\mathrm{regularization}}}$$
$$+ \underbrace{2 \sum_{i=1}^{L_{\mathrm{R}}^{-}} g_i(I_{\mathrm{no}}) - 2 \sum_{i=1}^{L_{\mathrm{R}}^{+}} g_i(I_{\mathrm{no}})}_{\mathcal{L}_{\mathrm{inflation}}}$$
$$+ \underbrace{\sum_{i=1}^{L_{\mathrm{W}}} g_i^2(I_{\mathrm{no}}) - 2 \sum_{i=1}^{L_{\mathrm{W}}^{-}} g_i(I_{\mathrm{no}}) + 2 \sum_{i=1}^{L_{\mathrm{W}}^{+}} g_i(I_{\mathrm{no}})}_{\mathcal{L}_{\mathrm{deflation}}} + L \Bigg] \tag{3}$$

The proof of Equation (3) is in Appendix A.1.

According to whether $g_i(I_{\mathrm{no}})$ is correctly decoded (i.e., $g_i(I_{\mathrm{no}}) \cdot M_i > 0$ ), we divide $g(I_{\mathrm{no}})$ into two parts with lengths $L_{\mathrm{W}}$ and $L_{\mathrm{R}}$. Additionally, we also divide $g_i(I_{\mathrm{no}})$ based on its sign (i.e., $g_i(I_{\mathrm{no}}) > 0$) into two parts with lengths $L^+$ and $L^-$. The symbols "W" and "R" represent "Wrong" and "Right", respectively, while "+" and "-" denote the positive and negative signs of $g_i(I_{\mathrm{no}})$, respectively. The combination of "W(R)" and "+(-)" represents the parts of $g(I_{\mathrm{no}})$ that simultaneously satisfy both conditions.

The objective of penalizing incorrectly decoding bits is completely allocated to the last three terms of Equation (3). To minimize the MSE loss, these three positive terms will converge to 0. Thus, we named them as the error deflation loss $\mathcal{L}_{\mathrm{deflation}}$. Minimizing $\mathcal{L}_{\mathrm{deflation}}$ by always outputting 0 for any input is a shortcut that may lead to model collapse,

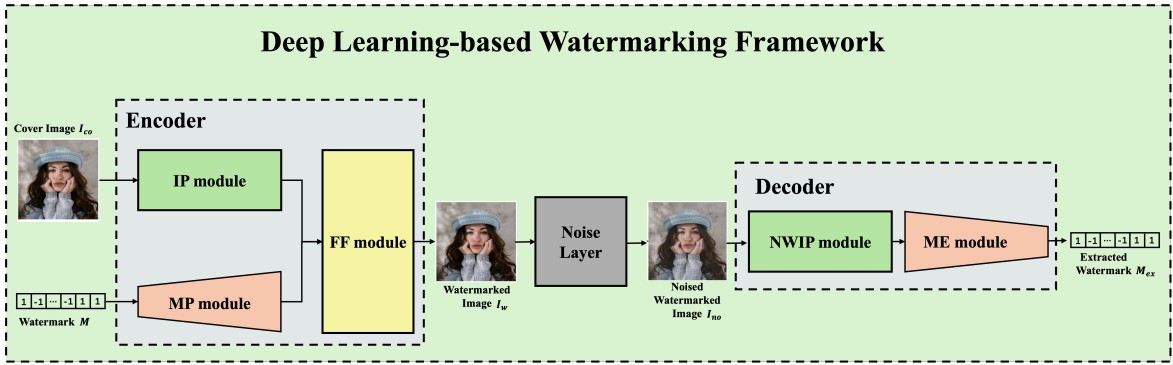

**Deep Learning-based Watermarking Framework**

*Figure 1.* The proposed deep learning-based framework comprises five modules. The encoder includes the image preprocessing (IP) module, the message preprocessing (MP) module, and the feature fusion (FF) module. The decoder contains the noised watermarked image preprocessing (NWIP) module and the message extraction (ME) module. A noise layer is also introduced between the encoder and decoder to distort the watermarked image into a noised version.

making it impossible to decide whether the output is correct or not. The two terms of $\mathcal{L}_{\text{inflation}}$ are both composed of correctly decoded parts ($L_{\text{R}}^{+(-)}$). Minimizing $\mathcal{L}_{\text{inflation}}$ does not directly reduce the decoding error; its primary function is to push correctly decoded $g_i(I_{\text{no}})$ away from the classification boundary 0, thus preventing model collapse. The final term, $\mathcal{L}_{\text{regularization}}$, also acts on the correctly decoded parts and serves as a complement to $\mathcal{L}_{\text{inflation}}$. It prevents unbounded growth of correctly decoded $g_i(I_{\text{no}})$, potentially leading to model output explosion. Among the three optimization directions, only $\mathcal{L}_{\text{deflation}}$ directly contributes to the goal of the information extraction task. While $\mathcal{L}_{\text{inflation}}$ and $\mathcal{L}_{\text{regularization}}$ within the MSE loss only play a positive role in stabilizing the training process, these additional optimization directions, inevitably occupy some model parameters, particularly limiting the performance of lightweight models.

## 3. Methodology

In this section, we first address the limitations of the surrogate loss identified in Section 2.2 from two perspectives to further explore the potential of lightweight models. Then, we introduce the roles of the five modules in the proposed new deep watermarking framework and illustrate the structure of a lightweight watermarking model built on this framework.

### 3.1. Mitigate the Gap between Two Objectives

**Detachable Projection Head** In this method, we continue to use the MSE loss, which ensures the stability of the model during training. However, during the training phase, we introduce an additional Detachable Projection Head to handle the redundant optimization directions that are unrelated to the decoding objective in the MSE loss. During inference, this Detachable Projection Head can be discarded to reduce

the model's parameter size, resulting in a lightweight model that can still decode correctly. The projection head's main function is normalization when considered separately from the backbone model. This means projecting the outputs of the backbone model into their corresponding label values. During training, the projection head does not care about the magnitude of the backbone model's outputs but requires these outputs to be distinguishable. If the backbone model's outputs are indistinguishable, the projection head cannot correctly project them, making it impossible to minimize the MSE loss. Therefore, when using MSE loss to optimize the overall model, the MSE loss forces the projection head's outputs to be more accurate to their label values. In turn, the projection head forces the backbone model's outputs to be more distinguishable by their label values, aligning well with the decoding goal. A detailed presentation of the distributions of the decoded messages from both the backbone model and the projection head is available in Appendix D.9.

The projection head consists of four identical projection blocks, and the structure of each projection block is shown in Appendix D.1. For the $j^{th}$ block, the input is $M_{\text{pex}}^{j}$, and the corresponding output is $M_{\text{pex}}^{j+1}$, which can be formulated as follows. The training and inference procedures of the detachable projection head method are illustrated in Fig. 2. More details can be found in Appendix C.

$$M_{\text{pex}}^{j+1} = A_j\left(M_{\text{pex}}^{j}\right) \otimes M_{\text{pex}}^{j} + B_j\left(M_{\text{pex}}^{j}\right) \quad (4)$$

where $\otimes$ denotes the dot product operation, and A and B are deep learning models. The total loss function $\mathcal{L}_{\text{PH}}$ can be represented as follows:

$$\mathcal{L}_{\text{PH}} = \lambda_1^{\text{PH}}\mathcal{L}_{\text{visual}} + \lambda_2^{\text{PH}}MSE\left(M_{\text{pex}}, M\right) \quad (5)$$

Here, $\lambda_1^{\text{PH}}$ and $\lambda_2^{\text{PH}}$ are weights to balance the trade-off between invisibility and robustness.

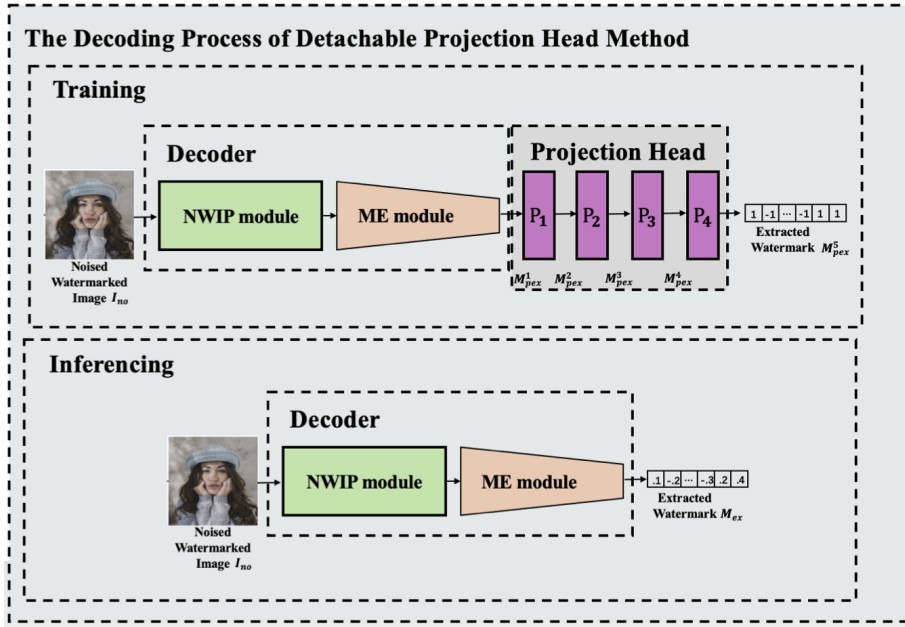

*Figure 2.* The decoding process of detachable projection head method.

**Decoding-Oriented Surrogate Loss** In this method, we refrain from using additional modules to stabilize training. Instead, we opt to mitigate the impact of irrelevant optimization directions from the MSE loss (for the implementation of BCE loss, please refer to Appendix D.2). $\mathcal{L}_{\text{deflation}}$ is directly responsible for reducing the decoding error by penalizing incorrectly decoded bits. The deflation loss in this method is adapted from $\mathcal{L}_{\text{deflation}}$ in Equation (3), which is formulated as follows:

$$\mathcal{L}_{\text{deflation}}\left(g(I_{\text{no}}), M\right) = \frac{1}{L}\mathbb{E}\left[\sum_{i=1}^{L_{\text{W}}^+} g_i(I_{\text{no}}) - \sum_{i=1}^{L_{\text{W}}^-} g_i(I_{\text{no}})\right] \tag{6}$$

We address the limitation in $\mathcal{L}_{\text{inflation}}$ by introducing a hyperparameter $\epsilon$, named "safe distance". This hyperparameter aims to mitigate the redundant impact of $\mathcal{L}_{\text{inflation}}$ on the model and prevent output collapse. We recognize that $\mathcal{L}_{\text{inflation}}$ encourages correctly decoded bits close to the decision boundary to move further away. However, it lacks a definition of "far enough", resulting in a broad influence on all correctly decoded bits, including those already sufficiently far from the boundary. To address this issue, we define a "safe distance" $\epsilon$. Only correctly decoded bits within a distance $\epsilon$ from the decision boundary are considered. Correctly decoded bits already more than $\epsilon$ away from the boundary are not subject to additional restrictions.

By limiting the influence of $\mathcal{L}_{\text{inflation}}$ to correctly decoded bits near the decision boundary, this approach reduces the unnecessary penalization of the model and prevents output collapse. Introducing the concept of a safe distance $\epsilon$ ensures the training stability while enhancing the performance of

the deep watermarking model. $\mathcal{L}_{\text{inflation}}$ in our method can be formulated as follows:

$$\mathcal{L}_{\text{inflation}}\left(g(I_{\text{no}}), M\right) = \frac{1}{L}\mathbb{E}\left[\sum_{i=1}^{L_{\text{R}}^- \cap L_{\text{R}}^{>-\epsilon}} g_i(I_{\text{no}}) - \sum_{i=1}^{L_{\text{R}}^+ \cap L_{\text{R}}^{<\epsilon}} g_i(I_{\text{no}})\right] \tag{7}$$

The total loss function $\mathcal{L}_{\text{DO}}$ can be represented as follows:

$$\mathcal{L}_{\text{DO}} = \lambda_1^{\text{DO}}\mathcal{L}_{\text{visual}} + \lambda_2^{\text{DO}}(\mathcal{L}_{\text{deflation}} + \mathcal{L}_{\text{inflation}}) \tag{8}$$

Here, $\lambda_1^{\text{DO}}$ and $\lambda_2^{\text{DO}}$ are weights to balance the trade-off between invisibility and robustness.

### 3.2. Proposed Framework

To delve deeper into the encoder and decoder, we propose a novel deep learning-based watermarking framework composed of five modules, as shown in Fig. 1. These modules provide a clearer functional division of the encoder and decoder, which has helped us find an efficient parameter allocation method when facing different distortions. We divide the encoder into three parts: **1) The image preprocessing (IP) module**, which aims to extract features from the original image comprehensively, facilitating subsequent feature fusion, such as the SE block (Hu et al., 2018) in MBRS (Jia et al., 2021) and De-END (Fang et al., 2022), or it transforms image features into the frequency domain to enhance robustness, as seen in CIN (Ma et al., 2022) using the Haar transform. **2) The message preprocessing**

**(MP) module's** primary task is to generate message features that align in shape with the image features generated by the image preprocessing module. HiDDeN (Zhu et al., 2018) employs a non-deep learning method by directly duplicating the messages. On the other hand, MBRS (Jia et al., 2021) and CIN (Ma et al., 2022) utilize deep learning methods based on transposed convolution layers. **3) The feature fusion (FF) module** integrates the message features and image features to produce the final watermarked images. StegaStamps (Tancik et al., 2020) adopts UNet-like model for multi-scale feature fusion, while CIN (Ma et al., 2022) and FIN (Fang et al., 2023) employ multiple invertible neural blocks for deep feature coupling. We divide the decoder into two parts: **4) The noised watermarked image preprocessing (NWIP) module** is the first module directly facing the noise layer. Its role is to mitigate the impact of distortions on the watermarked image and perform the initial extraction of message features from the noised watermarked images. **5) The message extraction (ME) module** is used to further extract the messages from the preprocessed features and reshape them to match the shape of the messages. In previous decoder architectures, these two modules were often deeply coupled, requiring uniform dimensions for the intermediate features. By decoupling, we can investigate the roles of these five modules separately and selectively remove non-essential modules to further compress the model. More detailed ablation experiments and analyses of the individual modules' impact on model performance are provided in Appendix D.3.

## 4. Experiments

In this section, we demonstrate the significant reductions in model size and computational complexity achieved by our proposed lightweight model compared to previous works. Additionally, we validate the effectiveness of our proposed PH and DO methods in improving the model's invisibility and robustness. In practice, during training, we employ a Combined Noise technique, where the model is exposed to a random noise layer in each mini-batch. This enables the model to learn robustness to multiple types of distortions at the same time. Additionally, during evaluation, we assess the model's robustness separately under different noise layers to demonstrate its general robustness. The Combined Noise Layer incorporates six different distortions: Gaussian Blur (GB) with a standard deviation of 2.0 and a kernel size of 7, Median Blur (MB) with a kernel size of 7, Gaussian Noise (GN) with a variance of 0.05 and a mean of 0, Salt & Pepper Noise (S&P) with a noise ratio of 0.1, JPEG Compression (JPEG) with a quality factor of 50, and Dropout (DP) with a drop ratio of 0.6. A more detailed experimental setup can be found in the Appendix D.

### 4.1. Model Size and Computational Complexity

To reduce the number of parameters and computational complexity, our proposed lightweight model only uses basic transposed convolution and convolution layers. The detailed model structure can be found in Appendix B.

*Table 1.* Comparison of parameter size and FLOPs with SOTA models. "Enc." and "Dec." denote the Encoder and Decoder, respectively. "Size (M)" refers to the number of parameters in millions (M), while "FLOPs (G)" indicates the computational complexity in billions of floating-point operations (GigaFLOPs).

| Method | Size (M) | | | FLOPs (G) | | |
|---|---|---|---|---|---|---|
| | Enc. | Dec. | Total | Enc. | Dec. | Total |
| CIN | 7.25 | 36.01 | 36.01 | 16.56 | 17.91 | 34.47 |
| MBRS | 0.56 | 20.24 | 20.80 | 8.38 | 6.77 | 15.15 |
| FIN | 0.75 | 0.75 | 0.75 | 1.78 | 1.78 | 3.56 |
| HiDDeN | 0.19 | 0.27 | 0.45 | 3.10 | 4.29 | 7.39 |
| Proposed | **0.009** | **0.007** | **0.016** | **0.15** | **0.07** | **0.22** |

**Model Size** The proposed lightweight model significantly reduces the number of parameters compared to state-of-the-art (SOTA) models, as shown in Table 1. Specifically, our model uses only 0.009M parameters for the encoder and 0.007M parameters for the decoder, resulting in a total of 0.016M parameters. This is a substantial reduction compared to the other models: CIN (36.01M), MBRS (20.80M), FIN (0.75M), and HiDDeN (0.45M). This drastic reduction in model size makes our model highly efficient in terms of storage requirements. The compact size is especially beneficial for deployment in resource-constrained environments such as mobile devices and embedded systems.

**Computational Complexity** The proposed model also achieves a remarkable reduction in FLOPs (Floating Point Operations), which is a critical measure of computational complexity. The FLOPs for our model are: Encoder (0.15G), Decoder (0.07G), Total (0.22G). In comparison, the FLOPs for the other models are significantly higher: CIN (34.47G), MBRS (15.15G), FIN (3.56G), and HiDDeN (7.39G). This indicates that our model not only requires fewer parameters but also operates with significantly lower computational complexity. This makes it suitable for real-time applications and scenarios where computational resources are limited

The proposed lightweight model excels in both storage efficiency and computational complexity, making it a highly practical solution for real-world applications. By significantly reducing the number of parameters and FLOPs, our model ensures that watermarking can be performed efficiently on devices with limited resources.

### 4.2. Invisibility and Robustness against Combined Noise

In Tables 2 and 3, the four watermarking models—HiDDeN, MBRS, CIN, and FIN—are implemented based on the au-

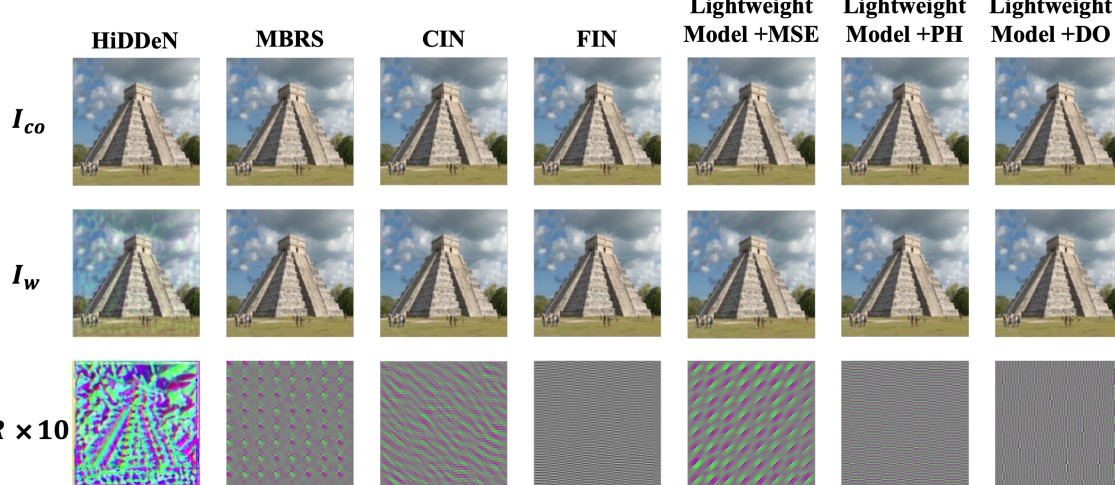

*Figure 3.* Visual comparison of watermarked images. Top: cover image. Middle: watermarked image. Bottom: the magnified difference $|I_w - I_{co}| \times 10$ between the cover image and the watermarked image.

*Table 2.* Benchmark comparisons on invisibility and robustness against combined noise. BCE, MSE, PH, and DO refer to the BCE loss, MSE loss, detachable projection head, and decoding-oriented surrogate loss.

| Method | PSNR↑ (dB) | Dropout (%) | JPEG (%) | GN (%) | S&P (%) | GB (%) | MB (%) | Ave (%) |
|---|---|---|---|---|---|---|---|---|
| HiDDeN | 27.28 | 74.59 | 74.63 | 77.36 | 77.81 | 77.17 | 75.28 | 76.14 |
| MBRS | 40.72 | 99.76 | 98.85 | 97.68 | 99.95 | 99.89 | 99.38 | 99.25 |
| CIN | 40.31 | 99.78 | 86.01 | 97.39 | **100** | 99.84 | 99.58 | 97.10 |
| FIN | 41.58 | 99.80 | **99.90** | 96.46 | 99.99 | 99.97 | 99.56 | 99.28 |
| Lightweight Model+BCE | 39.11 | 99.55 | 97.04 | 96.04 | 99.79 | 99.80 | 98.80 | 98.50 |
| Lightweight Model+MSE | 39.31 | 99.04 | 90.53 | **98.28** | 99.95 | 99.95 | 98.18 | 97.66 |
| Lightweight Model+PH | 41.67 | 99.99 | 98.92 | 97.21 | 99.99 | 99.96 | 99.59 | 99.28 |
| Lightweight Model+DO | **41.70** | **100** | 99.12 | 97.40 | **100** | **100** | **99.63** | **99.36** |

*Table 3.* Benchmark comparisons on invisibility and robustness against combined noise. KD (·) represents the proposed lightweight model trained using knowledge distillation, where the proposed model serves as the student model and (·) denotes the teacher model.

| Method | PSNR↑ (dB) | Dropout (%) | JPEG (%) | GN (%) | S&P (%) | GB (%) | MB (%) | Ave (%) |
|---|---|---|---|---|---|---|---|---|
| KD (HiDDeN) | 27.20 | 65.06 | 75.89 | 75.40 | 73.82 | 75.39 | 74.52 | 73.35 |
| KD (MBRS) | 38.98 | 83.40 | 88.17 | 87.82 | 93.42 | 92.70 | 89.05 | 89.09 |
| KD (CIN) | 38.90 | 65.75 | 85.03 | 86.86 | 90.29 | 86.98 | 82.92 | 82.97 |
| KD (FIN) | 40.10 | 96.04 | 96.25 | 95.76 | 97.96 | 97.60 | 96.39 | 96.67 |
| Lightweight Model+BCE | 39.11 | 99.55 | 97.04 | 96.04 | 99.79 | 99.80 | 98.80 | 98.50 |
| Lightweight Model+MSE | 39.31 | 99.04 | 90.53 | **98.28** | 99.95 | 99.95 | 98.18 | 97.66 |
| Lightweight Model+PH | 41.67 | 99.99 | 98.92 | 97.21 | 99.99 | 99.96 | 99.59 | 99.28 |
| Lightweight Model+DO | **41.70** | **100** | **99.12** | 97.40 | **100** | **100** | **99.63** | **99.36** |

thors' public code. For the proposed lightweight model, we used four different training methods: BCE loss, MSE loss, detachable projection head (PH), and decoding-oriented surrogate loss (DO).

**Benchmark with SOTA deep watermarking models** The benchmark comparisons in Table 2 demonstrate the performance of various watermarking models under combined noise. Due to space constraints, Table 2 uses PSNR as a representation of visual quality. For additional metrics related to visual quality, please refer to Appendix D.5. The comparison of performance under single noise with other models is reported in Appendix D.6. When using the original MSE or BCE losses, the proposed lightweight model shows inferior performance compared to other state-of-the-art (SOTA) large models such as CIN, FIN, and MBRS. This is evident from the average accuracy (Ave) values, which

*Table 4.* Benchmark comparisons on invisibility and robustness against diffusion-based attack.

| Method | t=0.03 | | t=0.05 | | t=0.1 | | t=0.2 | |
|--------|--------|--------|--------|--------|--------|--------|--------|--------|
| | PSNR(dB) | Acc(%) | PSNR(dB) | Acc(%) | PSNR(dB) | Acc(%) | PSNR(dB) | Acc(%) |
| **PRGAI Attack:** | | | | | | | | |
| MSE | 29.11 | 99.21 | 28.90 | 99.02 | 28.47 | 98.67 | 27.59 | 98.61 |
| PH | 29.22 | 99.90 | 29.11 | **99.70** | 28.52 | **99.31** | 27.66 | **99.51** |
| DO | **29.79** | **100** | **29.53** | 99.12 | **28.97** | 98.82 | **28.03** | 98.57 |
| **DiffPure Attack:** | | | | | | | | |
| MSE | 36.08 | 100 | 34.43 | 99.80 | 31.62 | 67.68 | 27.75 | 53.91 |
| PH | 36.21 | 100 | 34.48 | 99.80 | 31.64 | 69.24 | **27.86** | 56.26 |
| DO | **36.25** | **100** | **34.60** | **100** | 31.76 | **74.22** | 27.77 | **57.23** |

*Table 5.* Comparison of invisibility and robustness against combined noise by training with different components in MSE loss.

| Method | PSNR↑ (dB) | Dropout (%) | JPEG (%) | GN (%) | S&P (%) | GB (%) | MB (%) | Ave (%) |
|--------|------------|-------------|----------|--------|---------|--------|--------|---------|
| $L_{regularization}$ | **85.47** | 49.67 | 50.08 | 50.40 | 49.81 | 50.25 | 49.91 | 50.02 |
| $L_{inflation}$ | 5.16 | 49.78 | 49.78 | 49.78 | 49.78 | 49.78 | 49.78 | 49.78 |
| $L_{deflation}$ | 40.62 | 99.75 | 91.83 | 98.19 | 99.91 | 99.60 | 98.56 | 97.97 |
| MSE | 39.31 | 99.04 | 90.53 | **98.28** | 99.95 | 99.95 | 98.18 | 97.66 |
| DO | 41.70 | **100** | **99.12** | 97.40 | **100** | **100** | **99.63** | **99.36** |

serve as an overall indicator of performance against combined noise. Although the lightweight model reduces parameter and computational complexity, it fails to achieve the robustness of larger models when using MSE or BCE losses. This highlights the inherent limitations of the lightweight model when applied with the original MSE and BCE losses.

The introduction of the detachable projection head (PH) and decoding-oriented surrogate loss (DO) methods significantly enhances the performance of the proposed lightweight model in terms of both invisibility and robustness. Both PH and DO methods lead to a notable improvement compared to the MSE and BCE losses. PH Method: The detachable projection head improves the robustness and invisibility of the model without increasing the model size during inference. The average accuracy reaches 99.28%, which is higher than the MSE and BCE methods. DO Method: The decoding-oriented surrogate loss not only achieves the highest average accuracy of 99.36%, but also outperforms the other SOTA models including FIN and MBRS in terms of invisibility without adding any extra parameters or compromising efficiency. This demonstrates the effectiveness of the DO method in releasing the lightweight model's capability. The visual quality for these models is shown in Fig. 3.

**Benchmark with knowledge distillation method**  Table 3 illustrates the results obtained when training the lightweight model using knowledge distillation (Hinton et al., 2015), where our proposed lightweight model serves as the student model and larger models act as teacher models. It is observed that the performance of the student model, guided by knowledge distillation, tends to degrade compared to the

corresponding teacher model in Table 2 across various distortions. This degradation becomes more pronounced as the disparity in model size between the student and teacher models increases. This phenomenon underscores a limitation of knowledge distillation, where the effectiveness diminishes when there is a substantial gap in model capacities.

Among the student models, KD (FIN) performs the best, outperforming the lightweight model trained with MSE loss. This highlights the effectiveness of knowledge distillation in compressing model parameters and improving model efficiency. However, despite the improvements achieved by KD, the superiority of the PH and DO remains consistent. Even with higher invisibility, the proposed methods consistently outperform KD (FIN) across all single distortions. In summary, while knowledge distillation can compress model parameters and improve efficiency, the PH and DO methods consistently outperform it. The proposed methods exhibit remarkable effectiveness in enhancing the lightweight model's performance, making them valuable tools for optimizing lightweight watermarking models for real-world applications.

### 4.3. Invisibility and Robustness Against Diffusion-Based Attacks

Unlike incorporating distortions as a noise layer during training, we acknowledge that utilizing a diffusion model as a noise layer is impractical. However, the purification processes described in PRGAI (Zhao et al., 2023a) and DiffPure (Saberi et al., 2024) attacks both involve adding noise to the watermarked image, followed by multi-denoising steps. To simulate this type of attack, we employed a new composite

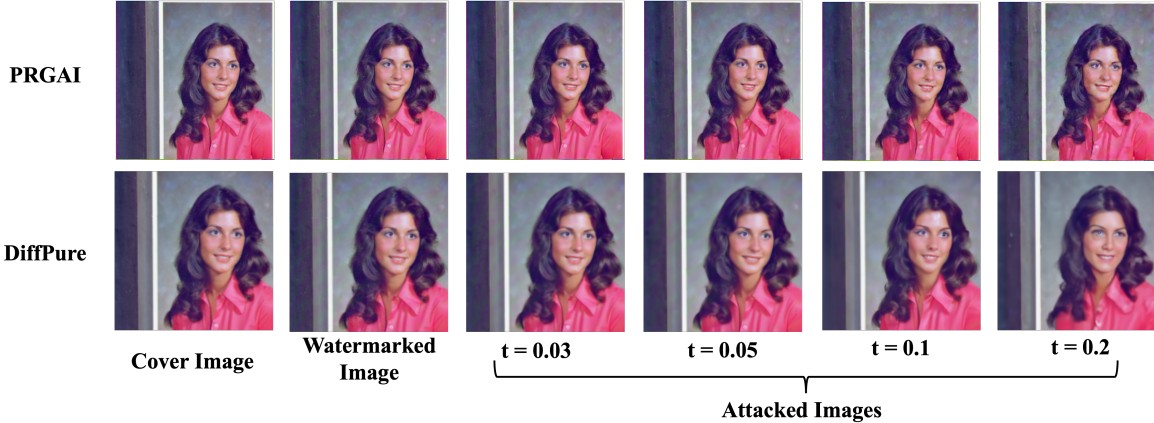

*Figure 4.* Visual comparison of images attacked by diffusion with different denoising time steps $t$.

noise layer: the watermarked images first pass through a Gaussian noise layer, followed by a median filter layer.

From Table 4, our model demonstrates nearly perfect robustness against the PRGAI attack. For the DiffPure attack, our results indicate that at $t = 0.05$, the accuracy of our DO method remains approximately 100%. At $t = 0.1$, the DO method still achieves 74.22% accuracy, despite a significant degradation in image quality after diffusion purification, which results in a PSNR drop of approximately 4 dB. While the semantic content of the attacked images remains intact, there is significant damage to the details (as shown in Fig. 4), rendering such high $t$ levels unacceptable for scenarios requiring detailed image preservation.

In these challenging conditions, our DO method retains 74.22% accuracy, demonstrating its robustness. Reducing the denoising steps $t$ helps preserve image details but also decreases the effectiveness of watermark removal. Thus, diffusion-based purification attacks have not yet fully evolved to eliminate watermarks without compromising image quality. Our method retains an advantage against these attacks by employing a composite noise layer of Gaussian noise and median filtering.

### 4.4. Ablation Study

**Impact of MSE components on model performance**  Table 5 presents the performance of the proposed lightweight model when trained using different components of the MSE loss: $\mathcal{L}_{\text{regularization}}$, $\mathcal{L}_{\text{inflation}}$, and $\mathcal{L}_{\text{deflation}}$. The results provide insights into the impact of each component on the model's performance in terms of invisibility and robustness against combined noise. The results demonstrate that using $\mathcal{L}_{\text{regularization}}$ or $\mathcal{L}_{\text{inflation}}$ in isolation does not effectively reduce the decoding error. $\mathcal{L}_{\text{regularization}}$ and $\mathcal{L}_{\text{inflation}}$ result in poor decoding accuracy, suggesting that these components do not contribute positively to the information extraction task and can lead to issues such as output collapse

or explosion. In contrast, training the model with $\mathcal{L}_{\text{deflation}}$ alone achieves better performance than the original MSE loss, particularly in robustness metrics. This improvement stems from the removal of irrelevant optimization directions, which allows the model to utilize previously consumed parameters more effectively. Despite these gains, the training process remains unstable, often requiring multiple adjustments to the weight of $\mathcal{L}_{\text{deflation}}$ to avoid model collapse. Additionally, while $\mathcal{L}_{\text{deflation}}$ improves robustness, the DO method still outperforms it in both visual quality and average accuracy, particularly for JPEG compression. In conclusion, while $\mathcal{L}_{\text{deflation}}$ shows potential in enhancing the lightweight model's performance, the stability and overall effectiveness of the DO method make it a superior choice for improving both invisibility and robustness in deep watermarking models.

## 5. Conclusion

This paper enhances the efficiency of deep learning-based watermarking models. We identify a mismatch between the commonly used decoding losses and the information extraction task. To address this, we propose the detachable projection head (PH) and decoding-oriented surrogate loss (DO), which reduce the impact of irrelevant components and improve model efficiency. We validate our methods by designing a lightweight model that achieves state-of-the-art visual quality and robustness against various distortions. Additionally, we introduce a five-module deep learning-based watermarking framework, providing a finer-grained division of the encoder and decoder. Extensive experimental results demonstrate the superiority of our proposed method over existing models in terms of invisibility, robustness, and efficiency. Moreover, the DO and PH methods are designed to be plug-and-play, making them widely applicable to lightweight models with different architectures.

## Acknowledgments

This work is funded by the NUS-NCS Joint Laboratory for Cyber Security (WBS A-0008542-00-00 & A-0008542-01-00).

## Impact Statement

This paper introduces a lightweight deep watermarking architecture designed to address the challenge of deploying high-performance watermarking models in resource-constrained environments. Our work improves the practical applicability of digital watermarking in real-world applications, particularly in scenarios such as video streaming, online education, and embedded systems, where computational power and storage are limited. By significantly reducing model size and computational demands, our approach facilitates the broader adoption of digital watermarking for intellectual property protection, ensuring effective copyright enforcement even on low-power edge devices. This can help mitigate unauthorized content distribution and strengthen digital rights management across various domains, including images, videos, and 3D content. Moreover, the computational efficiency of our method is well-aligned with energy-conscious system-on-chip (SoC) architectures, promoting sustainable AI deployment. Overall, our work advances practical, resource-efficient watermarking solutions while fostering responsible AI deployment in real-world applications. While our research may have broader societal implications, we do not identify any specific ethical concerns that require further discussion.

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

## A. Proof Results

### A.1. Equation (3) (restated). *MSE loss can be decomposed into seven terms as follows:*

$$MSE\left(g(I_{\mathrm{no}}), M\right) = \frac{1}{L}\mathbb{E}\left[\sum_{i=1}^{L_{\mathrm{W}}} g_i^2(I_{\mathrm{no}}) - 2\sum_{i=1}^{L_{\mathrm{W}}^-} g_i(I_{\mathrm{no}}) + 2\sum_{i=1}^{L_{\mathrm{W}}^+} g_i(I_{\mathrm{no}})\right.$$
$$\left. + 2\sum_{i=1}^{L_{\mathrm{R}}^-} g_i(I_{\mathrm{no}}) - 2\sum_{i=1}^{L_{\mathrm{R}}^+} g_i(I_{\mathrm{no}}) + \sum_{i=1}^{L_{\mathrm{R}}} g_i^2(I_{\mathrm{no}}) + L\right]$$

We divide the decoded bits $g_i(I_{\mathrm{no}})$ from the decoded message $g(I_{\mathrm{no}})$ into two groups based on whether they are correctly or incorrectly decoded, with lengths $L_{\mathrm{R}}$ and $L_{\mathrm{W}}$ respectively. Additionally, we can also divide these bits based on their sign into two groups with lengths $L^+$ and $L^-$. The combinations $R$ (right) and $W$ (wrong) with $+$ (positive) and $-$ (negative) represent the decoded bits that satisfy both conditions.

*Proof.* We expand the MSE loss in Equation (2) as follows:

$$MSE\left(g(I_{\mathrm{no}}), M\right) = \mathbb{E}\left[\frac{1}{L}\sum_{i=1}^{L}(g_i(I_{\mathrm{no}}) - M_i)^2\right] \tag{9}$$

$$= \frac{1}{L}\mathbb{E}\left[\sum_{i=1}^{L} g_i^2(I_{\mathrm{no}}) - 2\sum_{i=1}^{L} M_i g_i(I_{\mathrm{no}}) + \sum_{i=1}^{L} M_i^2\right] \tag{10}$$

The expectation in Equation (10) contains three terms, and the first term can be further decomposed into two more terms.

$$\sum_{i=1}^{L} g_i^2(I_{\mathrm{no}}) = \sum_{i=1}^{L_{\mathrm{W}}} g_i^2(I_{\mathrm{no}}) + \sum_{i=1}^{L_{\mathrm{R}}} g_i^2(I_{\mathrm{no}}) \tag{11}$$

The second term can be further decomposed into four more terms.

$$\sum_{i=1}^{L} M_i g_i(I_{\mathrm{no}}) = \sum_{i=1}^{L_{\mathrm{W}}^-} M_i g_i(I_{\mathrm{no}}) + \sum_{i=1}^{L_{\mathrm{W}}^+} M_i g_i(I_{\mathrm{no}}) + \sum_{i=1}^{L_{\mathrm{R}}^-} M_i g_i(I_{\mathrm{no}}) + \sum_{i=1}^{L_{\mathrm{R}}^+} M_i g_i(I_{\mathrm{no}}) \tag{12}$$

The four terms contain $g_i(I_{\mathrm{no}})$ as follows: $g_i(I_{\mathrm{no}})$ decoded incorrectly and with a negative sign, $g_i(I_{\mathrm{no}})$ decoded incorrectly and with a positive sign, $g_i(I_{\mathrm{no}})$ decoded correctly and with a negative sign, and $g_i(I_{\mathrm{no}})$ decoded correctly and with a positive sign. Therefore, the values of $M_i$ in these four groups are +1, -1, -1, and +1, respectively. Then, Equation (12) can be transformed as follows:

$$\sum_{i=1}^{L} M_i g_i(I_{\mathrm{no}}) = \sum_{i=1}^{L_{\mathrm{W}}^-} g_i(I_{\mathrm{no}}) - \sum_{i=1}^{L_{\mathrm{W}}^+} g_i(I_{\mathrm{no}}) - \sum_{i=1}^{L_{\mathrm{R}}^-} g_i(I_{\mathrm{no}}) + \sum_{i=1}^{L_{\mathrm{R}}^+} g_i(I_{\mathrm{no}}) \tag{13}$$

Since $M_i \in \{-1, 1\}$, the third term can be rewritten as follows:

$$\sum_{i=1}^{L} M_i^2 = L \tag{14}$$

Combining Equations (11), (13), and (14) yields:

$$\frac{1}{L}\mathbb{E}\left[\sum_{i=1}^{L_{\mathrm{W}}} g_i^2(I_{\mathrm{no}}) - 2\sum_{i=1}^{L_{\mathrm{W}}^-} g_i(I_{\mathrm{no}}) + 2\sum_{i=1}^{L_{\mathrm{W}}^+} g_i(I_{\mathrm{no}})\right.$$
$$\left. + 2\sum_{i=1}^{L_{\mathrm{R}}^-} g_i(I_{\mathrm{no}}) - 2\sum_{i=1}^{L_{\mathrm{R}}^+} g_i(I_{\mathrm{no}}) + \sum_{i=1}^{L_{\mathrm{R}}} g_i^2(I_{\mathrm{no}}) + L\right] \tag{15}$$

**A.2.** *BCE loss can be decomposed into four terms as follows:*

The decomposition process for BCE loss is similar to that of MSE loss. However, there are a few key points to note: 1) $M \in \{0, 1\}^L$; 2) The direct output of the decoder, $g_i(I_{\mathrm{no}})$, need to pass through the sigmoid activation function $\sigma(\cdot)$; 3) The classification boundary is no longer 0 but rather 0.5. Therefore, based on whether $\sigma(g_i(I_{\mathrm{no}}))$ is correctly decoded (i.e., $(\sigma(g_i(I_{\mathrm{no}})) - 0.5) \cdot (M_i - 0.5) > 0$), we divide $\sigma(g_i(I_{\mathrm{no}}))$ into two parts with lengths $L_{\mathrm{R}}$ and $L_{\mathrm{W}}$. We further categorize $\sigma(g_i(I_{\mathrm{no}}))$ based on its relationship with 0.5 (i.e., $\sigma(g_i(I_{\mathrm{no}}) > 0.5)$) into two parts with lengths $L^+$ and $L^-$.

$$
\begin{aligned}
BCE\left(\sigma(g(I_{\mathrm{no}})), M\right) = -\frac{1}{L}\mathbb{E}\Bigg[ & \underbrace{\sum_{i=1}^{L_{\mathrm{W}}^{-}} \log(\sigma(g_i(I_{\mathrm{no}}))) + \sum_{i=1}^{L_{\mathrm{W}}^{+}} \log(1 - \sigma(g_i(I_{\mathrm{no}})))}_{\mathcal{L}_{\mathrm{deflation}}} \\
& + \underbrace{\sum_{i=1}^{L_{\mathrm{R}}^{-}} \log(1 - \sigma(g_i(I_{\mathrm{no}}))) + \sum_{i=1}^{L_{\mathrm{R}}^{+}} \log(\sigma(g_i(I_{\mathrm{no}})))}_{\mathcal{L}_{\mathrm{inflation}}} \Bigg]
\end{aligned}
$$

The objective of penalizing incorrectly decoded bits is also totally reflected in the first two terms. To minimize the BCE loss, these terms encourage $\sigma(g_i(I_{\mathrm{no}}))$ to converge to the boundary at 0.5. The two components of $\mathcal{L}_{\mathrm{inflation}}$ are both comprised of correctly decoded parts ($L_{\mathrm{R}}^{+(-)}$). Minimizing $\mathcal{L}_{\mathrm{inflation}}$ does not directly reduce the decoding error; rather, its primary function is to push correctly decoded $\sigma(g_i(I_{\mathrm{no}}))$ away from the classification boundary at 0.5. Although $\mathcal{L}_{\mathrm{regularization}}$ is not explicitly included in the decomposition of BCE loss, the limitation on the unbounded growth of the model output $\sigma(g_i(I_{\mathrm{no}}))$ is implicitly enforced by the sigmoid activation function, as its output is constrained between 0 and 1.

*Proof.* The BCE loss is as follows:

$$
BCE\left(\sigma(g_i(I_{\mathrm{no}}))), M\right) = -\frac{1}{L}\mathbb{E}\left[\sum_{i=1}^{L} M_i \cdot \log(\sigma(g_i(I_{\mathrm{no}}))) + (1 - M_i) \cdot \log(1 - \sigma(g_i(I_{\mathrm{no}})))\right] \tag{16}
$$

$$
= -\frac{1}{L}\mathbb{E}\left[\sum_{i=1}^{L} M_i \cdot \log(\sigma(g_i(I_{\mathrm{no}}))) + \sum_{i=1}^{L}(1 - M_i) \cdot \log(1 - \sigma(g_i(I_{\mathrm{no}})))\right] \tag{17}
$$

The expectation in Equation (17) contains two terms, and the first term can be further decomposed into two more terms.

$$
\begin{aligned}
\sum_{i=1}^{L} M_i \cdot \log(\sigma(g_i(I_{\mathrm{no}}))) = & \sum_{i=1}^{L_{\mathrm{W}}^{-}} 1 \cdot \log(\sigma(g_i(I_{\mathrm{no}}))) + \sum_{i=1}^{L_{\mathrm{W}}^{+}} 0 \cdot \log(\sigma(g_i(I_{\mathrm{no}}))) \\
& + \sum_{i=1}^{L_{\mathrm{R}}^{-}} 0 \cdot \log(\sigma(g_i(I_{\mathrm{no}}))) + \sum_{i=1}^{L_{\mathrm{R}}^{+}} 1 \cdot \log(\sigma(g_i(I_{\mathrm{no}}))) 
\end{aligned} \tag{18}
$$

$$
= \sum_{i=1}^{L_{\mathrm{W}}^{-}} \log(\sigma(g_i(I_{\mathrm{no}}))) + \sum_{i=1}^{L_{\mathrm{R}}^{+}} \log(\sigma(g_i(I_{\mathrm{no}}))) \tag{19}
$$

The second term can also be further decomposed into two more terms.

$$
\sum_{i=1}^{L}(1 - M_i) \cdot \log(1 - \sigma(g_i(I_{\mathrm{no}}))) = \sum_{i=1}^{L_{\mathrm{W}}^{+}} \log(1 - \sigma(g_i(I_{\mathrm{no}}))) + \sum_{i=1}^{L_{\mathrm{R}}^{-}} \log(1 - \sigma(g_i(I_{\mathrm{no}}))) \tag{20}
$$

Combining Equations (19) and (20) yields:

$$-\frac{1}{L}\mathbb{E}\left[\sum_{i=1}^{L_{\mathrm{W}}^{-}}\log(\sigma(g_i(I_{\mathrm{no}})))+\sum_{i=1}^{L_{\mathrm{W}}^{+}}\log(1-\sigma(g_i(I_{\mathrm{no}})))\right.$$
$$\left.+\sum_{i=1}^{L_{\mathrm{R}}^{-}}\log(1-\sigma(g_i(I_{\mathrm{no}})))+\sum_{i=1}^{L_{\mathrm{R}}^{+}}\log(\sigma(g_i(I_{\mathrm{no}})))\right] \tag{21}$$

## B. Proposed Lightweight Model Structure

Here, we provide a comprehensive overview of the proposed lightweight model depicted in Fig. 5, which exclusively employs fundamental transposed convolution and convolution layers. The activation function utilized between each pair of layers is LeakyReLU.

In previous works, the partitioning of the encoder submodule has been well established, whereas our contribution lies in the delineation of the decoder. The separation of the two modules within the decoder is based on a specific criterion: layers with a stride of 1, which do not reduce the shape of the input, are designated as data processing layers, while layers with a stride greater than 2, which do reduce the input shape, are designated as data extraction layers. Previous approaches, such as those in MBRS and CIN, often mix or alternate these two kinds of layers. In our framework, however, these two kinds of layers are grouped into two distinct blocks, the noised watermarked image preprocessing (NWIP) module and the message extraction (ME) module. This division is primarily a structural separation. Since NWIP and ME are trained simultaneously and share the same objective function, a complete functional distinction between them is not feasible. However, the advantage of this structural separation is that the two modules do not need to share the same number of channels. This allows us to independently reduce the parameters of each part, enabling a more focused study and design of the decoder.

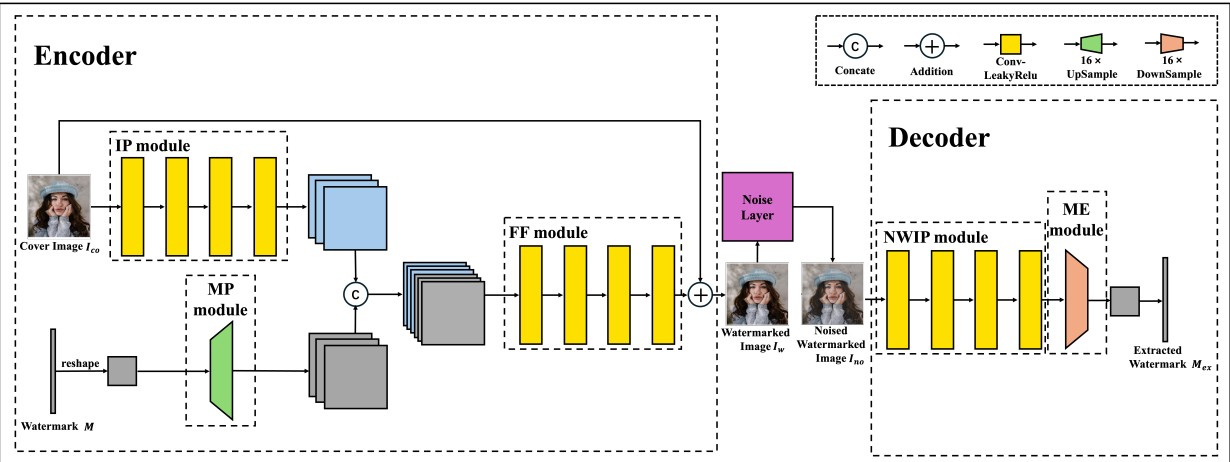

*Figure 5.* The structure of the proposed lightweight model. IP module represents image preprocessing module. MP module represents message preprocessing module. FF module represents feature fusion preprocessing. NWIP module represents noised watermarked image preprocessing module. ME module represents message extraction module.

## C. The Training and Inference Phases of PH

**Training Phase:** As illustrated in Fig. 2, the backbone network structure used in the Detachable Projection Head (PH) method is identical to that of the MSE loss-based backbone network, except for the addition of four identical projection blocks after the original decoder. Specifically, $M_{\mathrm{pex}}^{j}$ denotes the input to the $j^{th}$ Projection Block. Notably, $M_{\mathrm{pex}}^{1}$ is the output of the original backbone network, denoted as $M_{\mathrm{ex}}$.

For the decoder loss calculation, the MSE loss-based method uses the direct output $M_{\mathrm{ex}}$ from the backbone network. In contrast, the PH method utilizes the output of the last projection block, $M_{\mathrm{pex}}^{5}$ (since there are four projection blocks), for computing the MSE loss.

**Inference Phase:** As shown in Fig. 2, during inference, the PH method retains only the backbone network. We compute the decoding accuracy based on the output of the backbone network, $M_{\mathrm{ex}}$.

This section highlights how the PH method integrates the proposed lightweight structure during both the training and inference phases. The additional projection blocks in the training phase help refine the decoder output, while during inference, we simplify the structure to use only the backbone network for efficiency.

## D. Extensive Experimental Details and Results

**Datasets and Settings** All networks are trained on the COCO dataset (Lin et al., 2014) and tested on the classical USC-SIPI image dataset (Viterbi, 1977). The number of channels $C$, width $W$, and height $H$ of the images are set to 3, 128, and 128, respectively; the length $L$ of the secret message is set to 64. The safe distance $\epsilon$ in $\mathcal{L}_{\mathrm{DO}}$ is set to 0.1, and both $\lambda_1^{\mathrm{PH(DO)}}$ and $\lambda_2^{\mathrm{PH(DO)}}$ are initially set to 1. All experimental models are implemented through PyTorch (Collobert et al., 2011) and run on NVIDIA RTX 3090 (24GB). As for the optimizer, we used the Adam optimizer (Kingma & Ba, 2015) with a learning rate of 1e-3 and default hyperparameters. In the training phase, we apply a combination of seven types of distortions: Gaussian Blur (GB) with a standard deviation of 2.0 and a kernel size of 7, Median Blur (MB) with a kernel size of 7, Gaussian Noise (GN) with a variance of 0.05 and a mean of 0, Salt & Pepper Noise (S&P) with a noise ratio of 0.1, Dropout (DP) with a drop ratio of 0.6, JPEG Compression (JPEG) with a quality factor of 50, and JPEGSS (simulated differentiable JPEG distortion) with a quality factor of 50. These distortions are applied in combination to simulate a variety of noise conditions, ensuring the model can effectively handle different types of degradation during training.

**Benchmarks** To evaluate the efficiency, robustness, and invisibility of the proposed method, four widely used watermarking models are selected for comparison including HiDDeN (Zhu et al., 2018), MBRS (Jia et al., 2021), CIN (Ma et al., 2022) and FIN (Fang et al., 2023). For robustness testing, we choose six different distortions ("Gaussian Blur", "Median Blur", "Gaussian Noise", "Salt & Pepper Noise", "JPEG Compression" and "Dropout"), as well as a "Combined Noise" incorporating these six distortions.

**Metrics** For efficiency, we show the model size and the Floating Point Operations (FLOPs), where lower values signify higher efficiency. The peak signal-to-noise ratio (PSNR) is chosen to evaluate the visual quality of watermarked images. A larger PSNR value suggests smaller alterations from the original image, thus reflecting better invisibility. For robustness, decoding accuracy (ACC) is utilized as the metric. A higher ACC indicates better robustness.

### D.1. The Structure of Projection Block

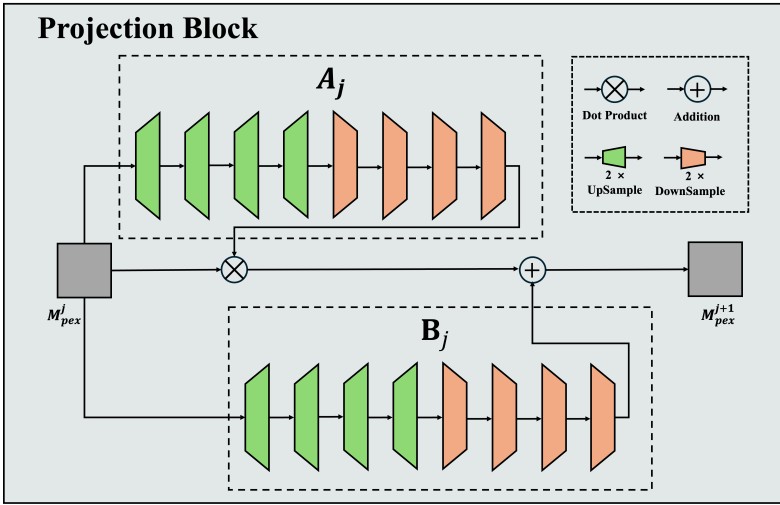

*Figure 6.* The structure of the projection block.

## D.2. Develop the Decoding-Oriented Surrogate Loss from BCE Loss

Similarly, the deflation loss here is adapted from $\mathcal{L}_{\text{deflation}}$ in Equation (21), which is formulated as follows:

$$\mathcal{L}_{\text{deflation}}^{\text{BCE}} = -\frac{1}{L}\mathbb{E}\left[\sum_{i=1}^{L_{\text{W}}^-}\log(\sigma(g_i(I_{\text{no}}))) + \sum_{i=1}^{L_{\text{W}}^+}\log(1-\sigma(g_i(I_{\text{no}})))\right] \tag{22}$$

$\mathcal{L}_{\text{inflation}}^{\text{BCE}}$ in here can be formulated as follows:

$$\mathcal{L}_{\text{inflation}}^{\text{BCE}} = -\frac{1}{L}\mathbb{E}\left[\sum_{i=1}^{L_{\text{R}}^{0.5<}\cap L_{\text{R}}^{>0.5-\epsilon}}\log(1-\sigma(g_i(I_{\text{no}}))) + \sum_{i=1}^{L_{\text{R}}^{0.5+\epsilon<}\cap L_{\text{R}}^{>0.5}}\log(\sigma(g_i(I_{\text{no}})))\right] \tag{23}$$

The total loss function $\mathcal{L}_{\text{DO}}^{\text{BCE}}$ can be represented as follows:

$$\mathcal{L}_{\text{DO}}^{\text{BCE}} = \lambda_1^{\text{DO}}\mathcal{L}_{\text{visual}} + \lambda_2^{\text{DO}}(\mathcal{L}_{\text{deflation}}^{\text{BCE}} + \mathcal{L}_{\text{inflation}}^{\text{BCE}}) \tag{24}$$

*Table 6.* Comparison of Performance between DO (MSE-based) and DO (BCE-based) Methods

| Method | PSNR↑ (dB) | Dropout (%) | JPEG (%) | GN (%) | S&P (%) | GB (%) | MB (%) | Ave (%) |
|---|---|---|---|---|---|---|---|---|
| DO (MSE) | 41.70 | 100 | 99.12 | 97.40 | 100 | 100 | 99.63 | 99.36 |
| DO (BCE) | 41.10 | 100 | 98.11 | 97.96 | 100 | 99.94 | 99.74 | 99.29 |

The comparative experiments for the DO method based on BCD loss are shown in Table 6. As can be seen, the performance of DO (MSE-based) and DO (BCE-based) are very similar.

## D.3. Impact of individual modules on model performance

*Table 7.* Benchmark comparisons on invisibility and robustness against different single distortions.

| Model | S&P Noise (%) | | | | Gaussian Noise (%) | | | |
|---|---|---|---|---|---|---|---|---|
| | PSNR(dB)↑ | r=0.08 | 0.09 | 0.1 | PSNR(dB)↑ | var=0.03 | 0.04 | 0.05 |
| w/o IP | 67.19 | **99.97** | 99.93 | 99.87 | **39.94** | 99.94 | 99.70 | 99.25 |
| w/o FF | 67.02 | 99.87 | 99.92 | 99.75 | 39.68 | **99.98** | 99.76 | 99.34 |
| w/o NWIP | 56.66 | 97.43 | 97.25 | 96.29 | 39.54 | 99.92 | 99.78 | 99.33 |
| Whole Model | **67.75** | **99.97** | **99.96** | **99.94** | 39.91 | 99.97 | **99.82** | **99.51** |

| Model | JPEG Compression (%) | | | | Dropout (%) | | | |
|---|---|---|---|---|---|---|---|---|
| | PSNR(dB)↑ | QF = 50 | 60 | 70 | PSNR(dB) ↑ | r=0.6 | 0.5 | 0.4 |
| w/o IP | 49.70 | 95.73 | 96.95 | 98.46 | 72.04 | 98.22 | 99.59 | 99.81 |
| w/o FF | 49.62 | 90.79 | 92.36 | 94.08 | 71.63 | 95.55 | 97.78 | 98.79 |
| w/o NWIP | 49.55 | 91.38 | 93.20 | 94.81 | 70.01 | 96.99 | 99.75 | 99.95 |
| Whole Model | **49.73** | **99.06** | **99.75** | **99.99** | **72.64** | **99.50** | **99.95** | **99.98** |

| Model | Gaussian Blur (%) | | | | Median Blur (%) | | | |
|---|---|---|---|---|---|---|---|---|
| | PSNR(dB)↑ | $\sigma = 0.5$ | 1 | 2 | PSNR(dB)↑ | w = 3 | 5 | 7 |
| w/o IP | 67.84 | 99.96 | 99.34 | **99.63** | 49.15 | 99.63 | 99.69 | 98.75 |
| w/o FF | 67.31 | 98.55 | 96.99 | 94.02 | 49.47 | 99.45 | 98.67 | 96.08 |
| w/o NWIP | 62.36 | 84.25 | 79.44 | 65.12 | 45.85 | 99.31 | 99.78 | 99.00 |
| Whole Model | **67.91** | **100** | **99.99** | 99.57 | **49.95** | **99.97** | **99.82** | **99.26** |

Fig. 1 shows a deep learning-based watermarking framework composed of five modules. Among them, the MP (Message Preprocessing) module and ME (Message Extraction) module are essential as they are directly responsible for the transformation and extraction of watermarks. Therefore, Table 7 focuses on the remaining three modules and examines their impact under various single distortions. The performance difference between the model without the IP (Image Preprocessing)

module and the whole model is minimal, indicating its limited contribution to the overall model performance. However, the roles of the FF (Feature Fusion) module and the NWIP (Noised Watermarked Image Preprocessing) module are consistently significant across various distortions. The absence of the NWIP module leads to severe degradation in performance under Gaussian blur and S&P noise. In summary, when model size is constrained, effective parameter allocation is crucial to handle different distortions in varying application scenarios. Our results highlight the importance of the FF and NWIP modules in maintaining robustness and invisibility of the watermark, while the IP module has a relatively minor impact.

### D.4. The Capacity of Lightweight Model

For testing the capacity of the lightweight model, the cover images are 3-channel color images with a width and height of $128 \times 128$. We follow the noiseless environment setting described in HiDDeN (Zhu et al., 2018), meaning there is no noise layer to distort the watermarked images $I_w$. Instead, $I_w$ is directly fed into the decoder for the accuracy test. The specific experimental results are shown in Table 8.

Table 8. The Capacity of Lightweight Model.

| Method | Message Length | Bits Per Pixel | PSNR(dB) | Accuracy(%) |
|---|---|---|---|---|
| **PH + Lightweight Model** | 64 | 0.0013 | 43.15 | 100 |
| | 256 | 0.0052 | 42.14 | 100 |
| | 1024 | 0.0208 | 41.11 | 100 |
| | 4096 | 0.0833 | 40.41 | 100 |
| | 16384 | 0.3333 | 39.12 | 100 |
| **DO + Lightweight Model** | 64 | 0.0013 | 42.91 | 100 |
| | 256 | 0.0052 | 42.14 | 100 |
| | 1024 | 0.0208 | 41.03 | 100 |
| | 4096 | 0.0833 | 40.12 | 100 |
| | 16384 | 0.3333 | 39.01 | 100 |

### D.5. Benchmark Comparisons on Visually Quality

Additional evaluations of the visual quality across the seven models under combined noise conditions were conducted. In addition to the previously discussed PSNR, four other metrics, including SSIM, LPIPS, $l_2$, and $l_{inf}$, are also incorporated. As shown in Table 9, the methods DO and PH consistently outperform the other models across all metrics, with the exception of SSIM, where they perform slightly worse than MBRS.

Table 9. Benchmark comparisons on visual quality base on five different metrics.

| Method | PSNR (dB)↑ | SSIM↑ | LPIPS↓ | $l_2 \downarrow$ | $l_{inf} \downarrow$ |
|---|---|---|---|---|---|
| HiDDeN | 27.28 | 0.87 | 0.062 | 85.12 | 0.38 |
| MBRS | 40.72 | **0.98** | 0.004 | 4.19 | 0.16 |
| CIN | 40.31 | 0.97 | 0.003 | 5.03 | 0.13 |
| FIN | 41.58 | 0.97 | 0.014 | 3.46 | 0.08 |
| Lightweight Model+MSE | 39.31 | 0.97 | 0.005 | 5.67 | 0.12 |
| Lightweight Model+PH | 41.67 | 0.97 | 0.002 | 3.46 | 0.07 |
| Lightweight Model+DO | **41.70** | 0.97 | **0.001** | **3.36** | **0.06** |

### D.6. Invisibility and Robustness against Single Noise

In Table 10, the proposed lightweight model, using the MSE loss, exhibits inferior performance compared to other SOTA large models (CIN, FIN, MBRS) under certain single distortions. Specifically, significant discrepancies are observed in Gaussian blur and median blur. Under Gaussian blur, the proposed lightweight model (MSE) shows lower invisibility and robustness compared to the MBRS, with a more pronounced difference observed under median blur, particularly when compared to the MBRS and CIN.

In contrast, our proposed methods, the detachable projection head (PH), and the decoding-oriented surrogate loss (DO),

show significant enhancements over the MSE loss. Notably, the PH and DO methods outperform MSE loss across almost all single distortions, showcasing higher PSNR and improved robustness. Noteworthy is the success of the DO method in enabling the proposed lightweight model to surpass other SOTA large models without increasing model parameters or sacrificing efficiency, demonstrating a significant advantage.

These findings underscore the considerable performance gains achievable by adopting PH and DO methods over traditional MSE loss in deep learning-based watermarking models, particularly in the context of lightweight models.

*Table 10.* Benchmark comparisons on invisibility and robustness against different single distortions. MSE, PH, and DO refer to the original MSE loss, detachable projection head, and decoding-oriented surrogate loss.

| Model | S&P Noise (%) | | | | Gaussian Noise (%) | | | |
|---|---|---|---|---|---|---|---|---|
| | PSNR(dB)↑ | r=0.08 | 0.09 | 0.1 | PSNR(dB)↑ | var=0.03 | 0.04 | 0.05 |
| HiDDeN | 31.93 | 96.81 | 96.73 | 96.69 | 26.57 | 87.64 | 87.31 | 87.14 |
| MBRS | 67.73 | 99.95 | 99.91 | 99.89 | 40.36 | 99.89 | **99.53** | 99.05 |
| CIN | 66.31 | 98.12 | 97.41 | 97.26 | 39.77 | **99.91** | 99.33 | 98.42 |
| FIN | 63.53 | 99.44 | 99.25 | 99.02 | 40.35 | 99.87 | 99.41 | 99.04 |
| MSE | 66.86 | 99.67 | 99.61 | 99.43 | 40.34 | 99.76 | 99.13 | 98.82 |
| PH | 67.34 | 99.96 | **99.97** | 99.86 | 40.16 | 99.81 | 99.38 | 98.73 |
| DO | **67.75** | **99.97** | 99.96 | **99.94** | **40.38** | 99.88 | **99.53** | **99.08** |

| Model | JPEG Compression (%) | | | | Dropout (%) | | | |
|---|---|---|---|---|---|---|---|---|
| | PSNR(dB)↑ | QF = 50 | 60 | 70 | PSNR(dB) ↑ | r=0.6 | 0.5 | 0.4 |
| HiDDeN | 24.68 | 78.72 | 79.58 | 79.86 | 30.31 | 86.65 | 86.69 | 86.84 |
| MBRS | 47.82 | 96.01 | 97.85 | 99.31 | 70.48 | 99.05 | 99.91 | 99.98 |
| CIN | 48.47 | 84.77 | 88.58 | 93.36 | 63.41 | 97.07 | 98.73 | 99.22 |
| FIN | 48.76 | 98.24 | 99.51 | **100** | 62.58 | 99.22 | 99.61 | 99.68 |
| MSE | 48.47 | 99.00 | 99.52 | 99.72 | 71.86 | 98.69 | 99.69 | 99.70 |
| PH | 49.68 | 99.02 | 99.55 | 99.85 | 71.27 | 99.43 | 99.85 | **100** |
| DO | **49.73** | **99.06** | **99.75** | 99.99 | **72.64** | **99.50** | **99.95** | 99.98 |

| Model | Gaussian Blur (%) | | | Median Blur (%) | | |
|---|---|---|---|---|---|---|
| | PSNR(dB)↑ | $\sigma = 1$ | 2 | PSNR(dB)↑ | w = 5 | 7 |
| HiDDeN | 29.02 | 81.76 | 60.65 | 34.12 | 79.28 | 75.03 |
| MBRS | 65.78 | 99.84 | 99.21 | 49.85 | 99.77 | **99.56** |
| CIN | 63.32 | 98.83 | 97.66 | 49.08 | 98.86 | 98.44 |
| FIN | 52.51 | 99.80 | 97.27 | 41.97 | 99.54 | 99.13 |
| MSE | 57.50 | 99.35 | 98.85 | 48.98 | 99.38 | 98.06 |
| PH | 65.47 | 99.34 | 98.55 | 49.84 | 99.75 | 98.93 |
| DO | **67.91** | **99.99** | **99.57** | **49.95** | **99.82** | 99.26 |

## D.7. Visual Quality under Combined Noise

In Fig. 3, we present the watermarked images $I_w$ embedded with watermarks by four different models, HiDDeN, MBRS, CIN, and FIN, as well as the lightweight models trained using three different methods when facing combined noise. It can be observed that not only do the watermark patterns differ significantly among the four different models, but also the watermark patterns generated by the lightweight models with the same architecture vary under different training methods. Although our methods, PH and DO, do not directly affect the encoder, since the encoder and decoder are trained together, the losses generated by PH and DO during backpropagation will also be propagated to the encoder, thereby influencing the watermark patterns. From Table 2, it can be seen that the lightweight models trained with PH and DO not only exhibit improved robustness but also achieve enhanced visual quality. Therefore, this influence is positive.

## D.8. The Effect of Discriminator on Proposed Lightweight Model

For the watermarking task, the discriminator serves as an additional module. In our work, to clearly demonstrate and validate the effectiveness of the proposed DO and PH methods, we chose to minimize the influence of other factors that could affect the model's visual quality and robustness. Therefore, we did not use the discriminator in our model.

To further illustrate the impact of including or excluding the discriminator on the visual quality and robustness of our

*Table 11.* Benchmark comparisons on invisibility and robustness with and without Discriminator (DIS).

| Method | PSNR↑ (dB) | Dropout (%) | JPEG (%) | GN (%) | S&P (%) | GB (%) | MB (%) | Ave (%) |
|---|---|---|---|---|---|---|---|---|
| DO w/o DIS | 41.70 | 100 | 99.12 | 97.40 | 100 | 100 | 99.63 | 99.36 |
| DO w DIS | 41.21 | 100 | 98.57 | 97.79 | 99.99 | 100 | 99.70 | 99.34 |
| PH w/o DIS | 41.67 | 99.99 | 98.92 | 97.21 | 99.99 | 99.96 | 99.59 | 99.28 |
| PH w DIS | 41.07 | 100 | 98.73 | 97.64 | 100 | 100 | 99.49 | 99.31 |

*Table 12.* Benchmark comparisons on visual quality base on five different metrics with and without Discriminator (DIS).

| Method | PSNR (dB)↑ | SSIM↑ | LPIPS↓ | $l_2 \downarrow$ | $l_{inf} \downarrow$ |
|---|---|---|---|---|---|
| DO w/o DIS | 41.70 | 0.97 | 0.001 | 3.46 | 0.06 |
| DO w DIS | 41.21 | 0.97 | 0.001 | 3.74 | 0.08 |
| PH w/o DIS | 41.67 | 0.97 | 0.002 | 3.36 | 0.07 |
| PH w DIS | 41.07 | 0.97 | 0.002 | 3.84 | 0.09 |

proposed lightweight model, we conducted extensive experiments using the method from MBRS (Jia et al., 2021). The specific experimental results are shown in Table 12 and Table 11. As seen, the presence or absence of the discriminator does not significantly affect the visual quality and robustness of our method.

*Table 13.* Comparison of Parameter Size and FLOPs Between the Lightweight Model and the Discriminator.

| Method | Size | FLOPs |
|---|---|---|
| Discriminator | 113.15K | 1.86G |
| Lightweight Model | 16.59K | 0.22G |

Furthermore, as shown in Table 13, the parameters and computational complexity of the discriminator are 6.8 times and 8.5 times larger than those of the entire lightweight model, respectively. To maintain the lightweight nature of the overall model, we also chose not to include the discriminator.

### D.9. In-Depth Analysis of the Detachable Projection Head (PH)

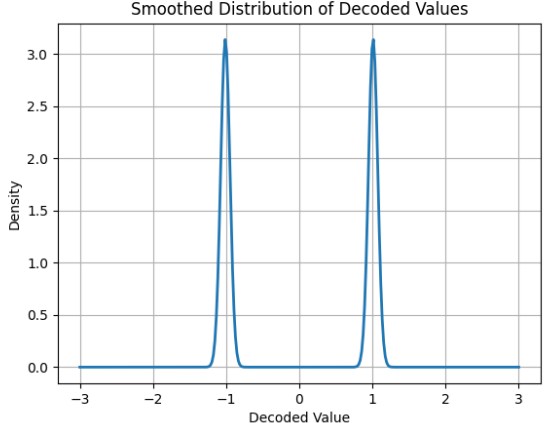

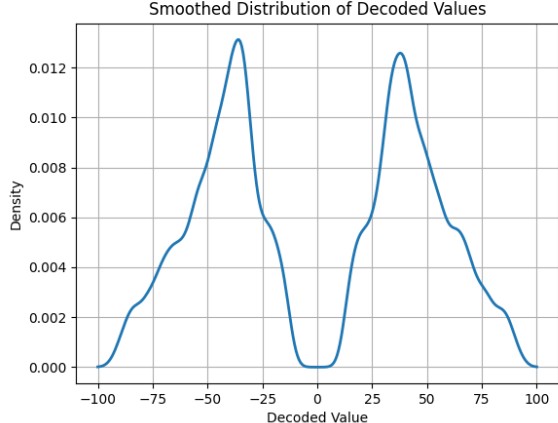

*Figure 7.* The smoothed distribution of decoded values for PH method with projection head.

*Figure 8.* The smoothed distribution of decoded values for PH method without projection head.

**The Difference Between the PH Method With and Without the Projection Head**  As illustrated in Fig. 7 and Fig. 8, the output distribution from the backbone model with projection head is densely centered around -1 and 1. This is expected, as the objective of the MSE loss is to push the outputs as close to the targets as possible. In contrast, the distribution of outputs decoded from the backbone model without projection head is more dispersed. This suggests that the projection head's main function is normalization when considered separately from the backbone model.

*Table 14.* Benchmark comparisons on invisibility and robustness with and without projection head.

| PH Blocks | PSNR (dB) | Dropout (%) | JPEG (%) | GN (%) | S&P (%) | GB (%) | MB (%) | Ave (%) |
|---|---|---|---|---|---|---|---|---|
| PH(w ph) | 40.74 | 99.98 | 98.43 | 98.07 | 99.94 | 100 | 99.14 | 99.26 |
| PH(w/o ph) | 40.74 | 100 | 98.62 | 98.03 | 99.99 | 100 | 99.54 | 99.36 |

**The Effect of Removing the Projection Head on Inference Performance**  Table 14 shows that discarding the projection head has a negligible impact on the model's performance during inference. As analyzed in the "Detachable Projection Head" section, the projection head mainly handles "normalization" which is not essential for the decoding goal. For decoding, only relative values are necessary.

*Table 15.* Influence of projection block numbers.

| PH Blocks | PSNR (dB) | Dropout (%) | JPEG (%) | GN (%) | S&P (%) | GB (%) | MB (%) | Ave (%) |
|---|---|---|---|---|---|---|---|---|
| 0 | 37.49 | 99.43 | 95.92 | 98.75 | 99.96 | 99.96 | 99.37 | 98.90 |
| 1 | 39.14 | 100 | 97.28 | 98.73 | 100 | 100 | 99.84 | 99.31 |
| 2 | 40.29 | 99.99 | 97.90 | 98.69 | 100 | 100 | 99.43 | 99.34 |
| 3 | 40.34 | 100 | 98.28 | 98.12 | 100 | 100 | 99.78 | 99.36 |
| 4 | 40.74 | 100 | 98.62 | 98.03 | 99.99 | 100 | 99.54 | 99.36 |
| 5 | 40.96 | 99.99 | 98.40 | 97.81 | 100 | 100 | 99.57 | 99.30 |

*Table 16.* Influence of channel numbers in projection block.

| Channel Numbers | PSNR (dB) | Dropout (%) | JPEG (%) | GN (%) | S&P (%) | GB (%) | MB (%) | Ave (%) |
|---|---|---|---|---|---|---|---|---|
| 4 | 40.44 | 99.09 | 93.88 | 97.58 | 99.98 | 99.92 | 98.54 | 98.17 |
| 8 | 40.56 | 98.66 | 94.49 | 98.11 | 100 | 99.81 | 98.73 | 98.30 |
| 16 | 41.43 | 100 | 97.22 | 97.75 | 99.99 | 100 | 99.50 | 99.09 |
| 32 | 41.67 | 99.99 | 98.92 | 97.21 | 99.99 | 99.96 | 99.59 | 99.28 |

**Impact of Block Number and Channel Number in the Projection Head on Inference Performance**  For the experiments involving the number of blocks, we fixed the channel number at 32. From Table 15, we observe a slight performance degradation as the block number decreases. Similarly, for the experiments with varying channel numbers, we fixed the block number at 4. As shown in Table 16, reducing the channel dimension also results in a slight decline in model performance.

From these two experiments, we can conclude that there is a trade-off between the size of the PH module and the performance of the lightweight model. Users can refer to these findings to select an appropriate PH size based on their computational resources and application requirements.

## D.10. Impact of $\lambda_1^{\text{PH(DO)}}$ and $\lambda_2^{\text{PH(DO)}}$

As shown in Table 17 and 18, initially, both and are set to 1 at the beginning of training. Throughout the training process, we increase the value of $\lambda_1^{\text{PH(DO)}}$ every 30 epochs. The tables below show the final values of $\lambda_1^{\text{PH(DO)}}$ and $\lambda_2^{\text{PH(DO)}}$, along with the corresponding visual quality and average accuracy of the model under combined noise. Our experiments indicate that $\lambda_1^{\text{PH(DO)}}$ and $\lambda_2^{\text{PH(DO)}}$ represent a trade-off between the visual quality of the encoder and the decoding accuracy

of the decoder. Larger values of lead to better visual quality but will result in reduced decoding accuracy. There is no one-size-fits-all standard for them. Users who prioritize the visual quality of the watermarked images and are willing to accept a reduction in accuracy might opt for higher values of $\lambda_1^{\text{PH(DO)}}$. Conversely, if decoding accuracy is more critical, lower $\lambda_1^{\text{PH(DO)}}$ values should be chosen.

*Table 17.* The influence of $\lambda_1^{\text{PH}}$ and $\lambda_2^{\text{PH}}$ for PH.

| Method | $\lambda_1^{\text{PH}}$ | $\lambda_2^{\text{PH}}$ | PSNR(dB) | Ave(%) |
|---|---|---|---|---|
| | 1 | 1 | 31.58 | 100 |
| | 10 | 1 | 36.36 | 99.85 |
| **PH** | 100 | 1 | 40.74 | 99.36 |
| | 1000 | 1 | 45.84 | 95.54 |
| | 10000 | 1 | 55.47 | 83.40 |

*Table 18.* The influence of $\lambda_1^{DO}$ and $\lambda_2^{DO}$ for DO.

| Method | $\lambda_1^{DO}$ | $\lambda_2^{DO}$ | PSNR(dB) | Ave(%) |
|---|---|---|---|---|
| | 1 | 1 | 27.58 | 100 |
| | 10 | 1 | 31.30 | 100 |
| **DO** | 1000 | 1 | 36.48 | 100 |
| | 100000 | 1 | 43.15 | 98.47 |
| | 1000000 | 1 | 49.03 | 93.16 |

## D.11. Impact of Safe Distance

In Table 19, we report the visual quality and decoding accuracy under combined noise with different safe distances $\epsilon$. Our experiments utilize the proposed lightweight model and the decoding-oriented surrogate loss (DO loss).

*Table 19.* The influence of safe influence $\epsilon$ in the decoding-oriented surrogate loss (DO loss).

| Safe Distance $\epsilon$ | PSNR↑ (dB) | Dropout (%) | JPEG (%) | GN (%) | S&P (%) | GB (%) | MB (%) | Ave (%) |
|---|---|---|---|---|---|---|---|---|
| 0.001 | 41.19 | 99.97 | 98.39 | 97.87 | 99.99 | 99.97 | 99.48 | 99.28 |
| 0.01 | 41.23 | **100** | 97.66 | 97.57 | **100** | 99.97 | 99.68 | 99.15 |
| 0.05 | 41.47 | **100** | 98.35 | 96.98 | 99.99 | **100** | **99.88** | 99.20 |
| **0.1** | **41.70** | **100** | **99.12** | 97.40 | **100** | **100** | 99.63 | **99.36** |
| 0.5 | 41.46 | 99.99 | 97.06 | 97.83 | 99.98 | **100** | 99.73 | 99.10 |
| 1.0 | 41.11 | **100** | 98.46 | 97.19 | 99.98 | **100** | 99.74 | 99.23 |
| 10.0 | 38.41 | 99.51 | 96.88 | **98.99** | **100** | **100** | 99.36 | 99.12 |

## D.12. Invisibility and Robustness against Geometric Distortions

Our work primarily aims to explore the feasibility of lightweight deep learning-based watermarking models. To this end, we validate the effectiveness and applicability of the proposed training methods, DO and PH, using a model with an intentionally simple structure and minimal parameters. We further investigate the broader applicability of these methods to different scenarios.

While our original lightweight model demonstrates strong robustness and visual quality against combined and multi-single distortions, achieving significant parameter reduction without compromising robustness across all distortion types remains inherently challenging. The distortions evaluated above are primarily digital channel-based distortions, which tend not to alter the geometric features of images significantly. Consequently, the substantial parameters and architectural components (e.g., the SE block in MBRS) designed for handling geometric distortions can be reduced without impacting performance in these cases. This characteristic explains why our original lightweight model, combined with the proposed DO and PH methods, performs effectively under such digital distortions.

*Table 20.* Benchmark comparisons on invisibility and robustness against geometric distortions, where RA represents RandomAffine, RP represents RandomPerspective, and RET represents RandomElasticTransform.

| Method | PSNR (dB) | RP (%) | RA (%) | RET (%) | Ave (%) |
|---|---|---|---|---|---|
| HiDDeN | 37.08 | 66.65 | 66.30 | 69.63 | 67.53 |
| MBRS | 48.09 | 98.05 | 98.44 | 99.71 | 98.73 |
| FIN | 42.05 | 73.63 | 74.45 | 99.51 | 82.53 |
| MSE with Lightweight Model | 39.89 | 82.22 | 82.34 | 99.41 | 87.99 |
| PH with Lightweight Model | 42.33 | 83.26 | 83.24 | 99.72 | 88.74 |
| DO with Lightweight Model | 43.66 | 84.23 | 84.69 | 99.68 | 89.53 |
| MSE with Lightweight Model + | 48.04 | 94.59 | 97.11 | 98.40 | 96.70 |
| PH with Lightweight Model + | 48.14 | 98.07 | 98.56 | 99.62 | 98.75 |
| **DO with Lightweight Model +** | **48.42** | **98.79** | **99.51** | **99.83** | **99.38** |

However, experiments in Table 20 reveal that our lightweight model's robustness declines when subjected to geometric distortions like RandomPerspective (RP), RandomAffine (RA), and RandomElasticTransform (RET). These distortions introduce greater complexity, requiring more sophisticated feature extraction to achieve optimal performance. While our original lightweight model does not achieve the best performance under geometric distortions, the DO method still ranks second, just behind MBRS.

To address this limitation, we extended the original lightweight model to enhance its robustness against geometric distortions. Specifically, we modified the noised watermarked image preprocessing (NWIP) module, the first module interacting with the noise layer, by incorporating an SE block and increasing the intermediate channel count from 12 to 32. The enhanced model, referred to as Lightweight Model +, achieves significant improvements in robustness to geometric distortions. Despite the enhancements, its total parameter count remains remarkably low at **0.056M**, representing only **12.44%** of HiDDeN, **7.47%** of FIN, **0.27%** of MBRS, and **0.16%** of CIN.

As shown in Table 20, the enhanced Lightweight Model + achieves substantial improvements across all geometric distortions. When coupled with the DO method, it attains the best performance in visual quality and robustness against all three distortions. The PH method also performs strongly, achieving comparable robustness, with only a slight gap in RandomElasticTransform, while outperforming MBRS in both visual quality and average accuracy.

In summary, the proposed training methods, DO and PH, exhibit broad applicability across various lightweight model architectures. We believe these methods will prove instrumental in advancing the development of lightweight watermarking models and will assist other researchers in achieving superior performance in this domain.

## E. Limitations

We propose two effective training methods: the detachable projection head (PH) and the decoding-oriented surrogate loss (DO). For PH, although we retain and use only the lightweight model during the inference stage, the training stage still requires additional storage space and computational resources to jointly train the detachable projection head, which is not efficient during training. For DO, while it does not require extra modules during training or inference, it introduces a new hyperparameter, the safe distance $\epsilon$. To achieve optimal performance, manual tuning of this hyperparameter is required, as demonstrated in Appendix D.11, which is not straightforward. Furthermore, as discussed in Appendix D.12, there is still room for improvement regarding geometric distortions. However, future researchers aiming to train lightweight models can readily adopt our training approaches to enhance model performance without modifying the underlying architecture. In conclusion, our work represents a significant advancement in balancing efficiency and robustness in watermarking models.

