# OpenReview forum: "Lightweight-Mark: Rethinking Deep Learning-Based Watermarking"
_ICML.cc/2025/Conference — ICML 2025 poster_

### Official Review · Reviewer_QY8P · 2025-03-13

**Overall Recommendation:** 3

**Summary:**

The paper proposes a deep watermarking framework that achieves state-of-the-art performance with significantly fewer parameters. The authors identify a mismatch between commonly used decoding losses and the decoding goal, which leads to parameter redundancy. To address this, they propose a Detachable Projection Head (PH) and a new loss function.

**Claims And Evidence:**

Yes

**Essential References Not Discussed:**

No

**Experimental Designs Or Analyses:**

No

**Methods And Evaluation Criteria:**

This scheme should add perceptual loss or adversarial training to enhance perceptual quality.

**Other Comments Or Suggestions:**

NO

**Other Strengths And Weaknesses:**

Strength:

There is a significant improvement in computation and efficiency. The proposed lightweight model utilizes less parameters compared to state-of-the-art frameworks, while achieving comparable or better performance in terms of invisibility and robustness.

Weaknesses:

(1)	This expansion of mean squared error seems to have been complicated. Can an effective analysis still be provided without considering the positivity or negativity of the decoding output?

(2)	This scheme should add perceptual loss or adversarial training to enhance perceptual quality.

**Questions For Authors:**

No

**Relation To Broader Scientific Literature:**

N/A

**Theoretical Claims:**

No

---

> ### Author Rebuttal · Authors · 2025-03-31
>
> **Weakness 1: Complexity of MSE Expansion and the Role of Positivity/Negativity**
>
> We appreciate the your insightful feedback. We understand your concern regarding the complexity.
>
> In watermark decoding tasks, positivity or negativity is a key criterion for determining whether decoding is correct. This not only serves as the foundation for decoding results but also plays an important role in our analysis, helping to express optimization behaviors such as **inflation** or **deflation**. Therefore, to ensure the rigor and completeness of the proof, considering positivity or negativity is indeed indispensable.
>
> However, your suggestion is very reasonable, and we plan to improve the paper in the following ways to enhance readability and clarity:
>
> **1. Simplification of Symbols and Formulae**:
>    To improve the readability of the formulas, we will simplify certain symbols and expressions. For example, we will rewrite $L_R^{+ and < \epsilon}$ as $L_R^{+} \cap L_R^{< \epsilon}$, making the expression more concise.
>
> **2. A More Intuitive Explanation of Equation 3**:
>    The goal of decoding is to ensure that the decoded bits are correctly separated from the boundary value (0), without considering the magnitude of each bit (e.g., 0.1 or 1). However, the objective of the MSE loss is not only to ensure that each bit is correctly separated from the boundary value but also to push the bits closer to ±1. Clearly, MSE loss is a stricter loss function, but this stricter requirement does not directly improve decoding accuracy. To highlight the gap between MSE loss and decoding accuracy, we have decomposed it into three parts:
>    - **L_deflation** mainly optimizes the portion of the decoding error and directly affects decoding accuracy.
>    - **L_inflation** and **L_regularization** optimize the portion of correct decoding but do not directly influence decoding accuracy.
>
> Additionally, based on the analysis of Equation 3, we propose two methods, **PH** and **DO**, to alleviate the redundancy introduced by **L_inflation** and **L_regularization**.
>
>
> ---
>
>
> **Weakness 2: Adding perceptual loss or adversarial training for perceptual quality**
>
> **1. Adding perceptual loss:**
>
> Following your suggestion, we expanded our experiments to include perceptual loss in our training process. To improve perceptual quality, we incorporated SSIM and LPIPS losses and present the results in Table 1.
>
> As shown in the Table 1, adding SSIM and LPIPS individually improves SSIM and LPIPS scores, particularly LPIPS, which **is reduced by at least half**.
>
> Although a slight drop in PSNR and average accuracy is observed, these trade-offs are reasonable given the enhanced perceptual quality.
>
> **Table 1. Benchmark comparisons on visual quality and robustness against combined noise**
>
> | **Method**         | PSNR | SSIM  | LPIPS   | Ave  |
> |--------------------|----------|-------|---------|---------|
> | PH                | 41.67    | 0.97  | 0.00184 | 99.28   |
> | PH + SSIM         | 41.14    | 0.98  | 0.00173 | 99.13   |
> | PH + LPIPS        | 40.14    | 0.97  | 0.00058 | 99.23   |
> | PH + SSIM + LPIPS | 40.06    | 0.98  | 0.00050 | 99.03   |
> | DO                | 41.70    | 0.97  | 0.00097 | 99.36   |
> | DO + SSIM         | 41.20    | 0.99  | 0.00158 | 99.30   |
> | DO + LPIPS        | 41.15    | 0.97  | 0.00047 | 99.31   |
> | DO + SSIM + LPIPS | 41.05    | 0.99  | 0.00044 | 99.06   |
> ---
>
> **2. Adding Adversarial Training：**
>
> To further investigate adversarial training, we analyzed the impact of adding a discriminator (DIS) to our proposed lightweight model. Detailed results are provided in Appendix E.8, and the key findings are summarized in Table 2.
>
> **Table 2. The Effect of the Discriminator (DIS) on the Proposed Lightweight Model**
>
> | **Method**    | **PSNR** | **SSIM** | **LPIPS** | **Ave** |
> |--------------|--------------|--------------|--------------|------------|
> | **PH w/o DIS**  | 41.67           | 0.97       | 0.00184       | 99.28       |
> | **PH w DIS**    | 41.07           | 0.97       | 0.00157       | 99.31       |
> | **DO w/o DIS**  | 41.70           | 0.97       | 0.00097       | 99.36       |
> | **DO w DIS**    | 41.21           | 0.97       | 0.00062       | 99.34       |
>
> As observed, the discriminator does indeed lead to some improvement in perceptual quality (LPIPS). Additionally, **efficiency is also a key metric for our model**. As shown in Appendix E.8 Table 13, the discriminator requires **6.8×** more parameters and **8.5×** more FLOPs than our entire lightweight model. Given the small perceptual gain and the substantial increase in computational cost, we prioritize **perceptual loss** as a more efficient approach to enhancing perceptual quality.
>
> ---
>
> We sincerely appreciate your insightful suggestion regarding perceptual quality. Your recommendation to incorporate **perceptual loss** has proven to be effective and efficient, and we will include this experimental analysis in the final version of our paper.

---

### Official Review · Reviewer_R1ca · 2025-03-15

**Overall Recommendation:** 3

**Summary:**

This manuscript presents a novel approach to deep learning-based watermarking, aiming to balance efficiency, invisibility, and robustness while reducing computational cost. The key contributions include: 1. Decoding-Oriented Surrogate Loss (DO): The authors identify a mismatch between commonly used decoding losses (e.g., MSE and BCE) and the actual decoding objective. They propose a surrogate loss that mitigates the influence of optimization directions unrelated to decoding accuracy. 2. Detachable Projection Head (PH): A temporary module introduced during training to handle decoding-irrelevant optimization directions, which is removed during inference to reduce model complexity. 3. Lightweight Model: The proposed watermarking model achieves state-of-the-art robustness and invisibility while using only 2.2% of the parameters of previous frameworks. 4. Comprehensive Experiments: The authors validate their approach across multiple distortions (Gaussian blur, noise, JPEG compression, etc.) and show robustness against diffusion-based watermark removal attacks.

**Claims And Evidence:**

The manuscript claims that existing decoding losses introduce parameter inefficiency by optimizing directions that do not directly contribute to decoding accuracy. To address this, it proposes the Decoding-Oriented Surrogate Loss (DO), which aims to eliminate irrelevant optimization directions, and the Detachable Projection Head (PH), a temporary module used during training to handle decoding-irrelevant tasks. The paper further claims that these methods enhance both robustness and invisibility while significantly reducing model complexity. Experimental results are presented to support these claims, demonstrating that the proposed methods achieve improved decoding accuracy and visual quality across multiple distortions while using only a small fraction of the parameters.

**Essential References Not Discussed:**

-

**Experimental Designs Or Analyses:**

1. Soundness: The experiments are well-structured, testing both traditional and diffusion-based watermark removal techniques. Additionally, the authors compare their method with knowledge distillation techniques, demonstrating the effectiveness of their approach.
2. Ablation Studies: The paper provides detailed analyses of different loss components (Table 5), verifying the necessity of each proposed module. The five modules in the proposed watermarking framework are also thoroughly tested in Table 7 under different distortions.
However, as mentioned in the limitations, the choice of some important hyperparameters is not sufficiently clear. For example, the safe distance (\epsilon) in the DO loss is manually set. Could this parameter be improved with automated tuning to enhance performance and ease of use?

**Methods And Evaluation Criteria:**

- Baseline Comparisons: The paper benchmarks against multiple SOTA watermarking models (HiDDeN, MBRS, CIN, FIN).
- Distortion Types: Assesses performance under Gaussian blur, noise, JPEG compression, dropout, salt & pepper noise, and diffusion-based attacks.
- Metrics: Uses PSNR, SSIM, LPIPS (visual quality), accuracy (robustness), and FLOPs/parameter numbers (efficiency).

**Other Comments Or Suggestions:**

1. Clarification on Hyperparameters:
The manual setting of certain hyperparameters, such as the safe distance (ϵ) in the DO loss, needs further clarification. How is a reasonable value for ϵ chosen in practice? Also, could you explore ways to automate the optimization of ϵ during training to improve adaptability and ease of use?
2. Figures in Appendix:
Figures like Figure 3 in Appendix B and Figure 4 in Appendix C are crucial for understanding the methods. I suggest moving these figures to the main body of the manuscript to improve accessibility and clarity.

**Other Strengths And Weaknesses:**

Strengths:
- The approach is interesting to the digital watermarking community, offering a simple and easy-to-deploy method. The experimental validation is comprehensive, covering multiple distortions and diffusion-based attacks.
- The proposed separable projection head (PH) and decoding-oriented alternative loss (DO) are effective in mitigating the negative impact of irrelevant optimization directions, enabling the lightweight model to achieve state-of-the-art performance.
- The novel lightweight watermarking framework outperforms existing models in terms of invisibility, robustness, and efficiency, making it a valuable solution for applications with limited computational resources.
Weakness:
- As mentioned in the limitations section, the safe distance (\epsilon) in the DO loss is manually set. Why hasn't an automated tuning approach been considered for this parameter? Wouldn't this enhance both performance and ease of use, making the method more robust and adaptable?
- I understand that due to space limitations, the authors have placed a lot of content in the Appendix. However, some important content should be in the main text. For example, Figure 3 in Appendix B and Figure 4 in Appendix C are essential visual representations that are crucial for the paper's presentation and should have been incorporated into the main body of the manuscript.

**Questions For Authors:**

1.	What motivates the design of the lightweight deep watermarking model?
2.	The manuscript uses several evaluation metrics like PSNR, SSIM, and LPIPS to evaluate visual quality. Can you provide more detailed insights on how these metrics compare in terms of their correlation with human perception of image quality, and whether the proposed method could be further improved in subjective visual quality?

**Relation To Broader Scientific Literature:**

The main problem addressed by this paper is still the visual quality and robustness of deep learning-based watermarking models (e.g., HiDDeN, CIN, FIN, MBRS). Beyond this, the authors also consider model lightweighting, setting their work apart from knowledge distillation methods (e.g., Hinton et al., 2015; Zhao et al., 2022), as their approach does not require a teacher model. Instead, the focus is primarily on improving the shared issues present in the widely used decoding losses (MSE and BCE loss) within watermarking tasks.

**Theoretical Claims:**

The manuscript provides a mathematically decomposition of MSE loss (Equation 3) and BCE loss (Equation 21, Appendix A.2), clearly identifying components that do not directly contribute to decoding accuracy. These insights form the foundation for the proposed Decoding-Oriented Surrogate Loss (DO), which offers an innovative approach to improving efficiency by focusing optimization on decoding-relevant directions. This finding is interesting to the deep learning based watermarking community.

---

> ### Author Rebuttal · Authors · 2025-03-31
>
> Thank you for your thorough review and the constructive comments on our manuscript. Below, we address your comments to clarify and improve our manuscript:
>
> **Weakness 1 & Suggestion 1: Can ($\epsilon$) in DO be Automated Tuning and How to Choose a Reasonable ($\epsilon$)?**
>
> **1. Can Safe Distance ($\epsilon$) in DO be Automated Tuning?**
>
> Following your suggestion, we attempted to make $\epsilon$ a learnable parameter integrated into the optimization process. We initially set $\epsilon$ to 0.1 and left other training settings unchanged. During training, we monitored both decoding accuracy (ACC) and the value of $\epsilon$. We found that $\epsilon$ decreased continuously as training progressed and eventually reached 0, leading to instability.
>
> At this point, the ACC dropped to around 50%, and the model was unable to decode correctly. We analyzed this behavior and identified that in the DO method, the role of $\epsilon$ in the DO method is to penalize values close to the boundary (0). A larger $\epsilon$ applies a broader penalty, while a smaller $\epsilon$ shrinks the penalty range. When $\epsilon$ reaches 0, no penalty is applied, and the model effectively minimizes $L_{inflation}$ by setting $\epsilon = 0$, reducing the DO method (i.e., $L_{deflation} + L_{inflation}$) to $L_{deflation}$. As shown in Section 2.2 ("The Gap between Two Objectives") of our original paper, this causes model instability, which we further validated and reported in Table 5 of Section 4.4 (Ablation Study).
>
>  **How to Choose a Reasonable ($\epsilon$)?**
>
> We acknowledge that directly incorporating $\epsilon$ as a learnable parameter was not feasible. However, your suggestion was valuable, and we conducted extensive experiments on the model's performance under varying $\epsilon$ values. The results of these experiments are presented in Appendix E.11, offering guidance on selecting an appropriate $\epsilon$.
>
> For now, we empirically selected $\epsilon$ based on these experiments, balancing model stability and optimal performance.  We also plan to explore more automated and convenient methods in future research. We hope this explanation clarifies the challenges we faced and addresses your concerns.
>
> ---
>
> **Weakness 2 & Suggestion 2: More Visual Results in Main Text.**
>
> Thank you for your constructive comment! Indeed, due to the current length constraints, we are unable to include more images in the main text now. However, in the final version, we will incorporate Figures 3 and 4 into the main body of the paper.
>
> ---
> **Question 1: The Motivation of the lightweight deep watermarking model.**
>
> The motivation behind the lightweight deep watermarking architecture stems from the practical demands of deploying models in resource-constrained environments. The motivations are as follows:
>
> **1. Practical Need for Lightweight Models in Real-World Applications**
> Digital watermarking plays a crucial role in protecting intellectual property across various domains, such as images, videos, and 3D content. However, high-performance models are often impractical for deployment due to their:
>
> - **Large parameter sizes**, which increase storage requirements.
> - **High computational demands**, which exceed the capacity of many real-world systems.
>
> This challenge is particularly acute in scenarios like **video streaming** and **online education**, where devices must operate efficiently within strict computational and energy constraints. High-performance lightweight models are essential to enable such deployment while remaining effective for copyright protection.
>
> **2. Importance of Lightweight Design in SoC Architectures**
> In SoC-based environments, lightweight design is not just beneficial but imperative:
>
> - **Storage Constraints**: The storage capacity in embedded devices is limited, making parameter-efficient models essential.
> - **Computational Efficiency**: SoCs lack significant computational power. Lightweight models reduce energy consumption, enabling practical deployment on edge devices.
>
> ---
> **Question 2: Further Improving Subjective Visual Quality.**
> - **PSNR** primarily measures pixel-level differences but does not reflect visual quality well in terms of image structure and texture details.
> - **SSIM** considers luminance, contrast, and structure, offering a better alignment with human perception, but it still does not fully capture complex visual features.
> - **LPIPS** compares feature representations from deep neural networks, providing a closer approximation to human perception, especially in terms of high-level perceptual quality.
>
> For improving subjective visual quality, more detailed experiments can be found in weakness 2 of Reviewer QY8P's feedback. For your convenience, we summarize the results here. We experimented with two approaches: directly adding SSIM and LPIPS losses, and using a discriminator. Both methods effectively improved subjective visual quality (LPIPS), but adding SSIM and LPIPS losses was computationally more efficient.

---

### Official Review · Reviewer_6BgQ · 2025-03-23

**Overall Recommendation:** 3

**Summary:**

The authors address the challenges of computational efficiency and accuracy in steganography. By identifying shortcomings in commonly used decoding loss functions, such as MSELoss and BCELoss, they introduce two techniques to mitigate these issues.

The first method, Detectable Projection Head (PH), scales the decoding output to maintain values close to 1 (or -1), improving training stability. The second method, Decoding-Oriented Surrogate Loss (DO), selectively filters out loss contributions from correctly decoded bits, focusing only on incorrectly decoded bits and those near the decision boundary. This targeted approach stabilizes training and enhances decoding accuracy.

Furthermore, the authors demonstrate that their approach achieves similar or slightly better performance than the previous state-of-the-art method, FIN, while enabling the use of a smaller model, thereby reducing computational overhead without sacrificing effectiveness.

### Update after Rebuttal

I would like to revise my score from 2 to 3, as the extended experimental results show that PH (referring to Table 1 in the second-round response) and DO (referring to Table 2 in the second-round response) offer valuable contributions to the development of future watermarking frameworks. Additionally, all questions and suggestions raised during the two-round discussions have been adequately addressed. This work demonstrates the potential for producing robust watermarking techniques while maintaining a lightweight model.

**Claims And Evidence:**

Yes. The authors provide the convincing experimental result that support their claims.

The authors compare their method's performance with prior works, demonstrating its effectiveness in terms of lightweight design and decoding accuracy. They also validate the soundness of their loss decomposition and grouping through an ablation study.

**Essential References Not Discussed:**

I did not identify any relevant works that are not referenced in the paper.

**Experimental Designs Or Analyses:**

The authors compare message decoding performance against four baselines: HiDDeN, MBRS, CIN, and FIN. The proposed method (lightweight model + DO) achieves performance comparable to the state-of-the-art (SOTA) method FIN, with slight improvements in visual preservation and decoding accuracy.

**Methods And Evaluation Criteria:**

The Decoding-Oriented Surrogate Loss (DO) method makes sense for the problem.

For evaluation, the authors use an image dataset with a resolution of 3 × 128 × 128, embedding watermark messages (default L = 64) and assessing extraction accuracy across various noisy channels. The evaluation metrics, including the visual preservation score (PSNR) and decoding accuracy, are standard metrics commonly used in the field. Note that they also evaluate robustness against two watermark removal attacks, PRGAI and DiffPure, demonstrating the effectiveness of the watermark under adversarial cases.

**Other Comments Or Suggestions:**

I believe that the subscript should be written in normal text—for instance, $L_{\text{deflation}}$ rather than $L_{deflation}$—when it represents a word. Moreover, using $\log$ instead of $log$ is much more preferred. To express the condition "and" in the formula more clearly, it would be better to use the intersection operator—such as $L^{−}_R \cap L^{>−ε}_R$— rather than $L^{−\text{and}>−ε}_R$.

Some typos were found in the paper:

1. L159, second column: The phrase "... first three terms of Equation (3) ..." seems incorrect, as this paragraph discusses the loss contributed by wrongly decoded bits. It should likely read "the last three terms."
2. L359, second column: The citation is missing the year.

**Other Strengths And Weaknesses:**

**Strengths**
- The paper is well-written and easy to follow overall.

**Limitations**
- The authors address limitations in the appendix: there is still room for improvement in the robustness of the watermarking against geometric distortion channels.

**Weakness**
- Some details remain unclear. Please refer to the Questions section.

**Questions For Authors:**

1. Is it possible to bound the output value of the decoder to the range [0, 1] using the $\text{tanh}$ function when using MSELoss? Currently, without the projection head, the output ranges from [-100, 100]. However, for BCELoss, a sigmoid function is typically added to restrict the co-domain of the decoder. Would applying a $\text{tanh}$ function with MSELoss produce a similar effect, or is there a specific reason for using the current approach?
2. I am confused about the term "Combined noise," as the performance of these perturbations is evaluated and reported separately. Please explain this term and the experiment design in more detail.
3. Referring to Section 4.3, the authors use Gaussian noise and a median filter as surrogates for diffusion-based watermark removal attacks for some reasons. Have you tried using a few-step diffusion model or Tweedie Estimation to avoid the multi-step sampling?
4. Referring to Table 4, the authors only compared the robustness of the baseline (MSE) and the proposed methods (PH and DO). What about prior works (HiDDeN, MBRS, CIN and FIN)?
5. Referring to Figure 5, it would be better to also show the watermarked image without an attack.
6. Referring to Table 10, the authors show the robustness when adjusting the noise layer with *less* perturbation. What about when the noise layer is adjusted with *more* perturbation?

**Relation To Broader Scientific Literature:**

I am not familiar with the broader scientific literature in this area, so I am unable to provide a detailed answer to this question.

**Theoretical Claims:**

I reviewed Appendix A and E.2, where the authors demonstrate how the loss can be decomposed and grouped based on decoding correctness (Right/Wrong) and decoded value (+/-). The mathematical formulation appears sound, and no major issues were identified in the proofs.

---

> ### Author Rebuttal · Authors · 2025-04-01
>
> **Notation and Typographical Corrections**
>
> Thank you for your detailed review. We will ensure proper formatting of subscripts and expressions in the final version.
>
> Regarding the typos, you are right that in L159 it should be "the last three terms," and the citation in L359 should be "2024". We will address these issues in the final version.
>
> ---
>
> **To provide a comprehensive response, we have included some results in the anonymous link:**
>
> https://anonymous.4open.science/r/Rebuttal-ICML-8393/Rebuttal_ICML2025_%208393.pdf
>
> **Q1:** Thank you for your constructive suggestion! If we remove the projection head, we could indeed add a Tanh function after the decoder to constrain the output to the range (-1,1), as shown in Figure 3 (linked). However, we do not impose such a restriction on the decoder output because the success of decoding depends on the relationship between the decoded bits and the boundary value (0 for MSELoss, 0.5 for BCELoss), rather than on the magnitude of the output itself. For example, an output of 0.1 and 2 both indicate the same decoded bit '1' under MSELoss.
>
> Adding a Tanh function would not alter the relative relationship between the decoded bits and the boundary (i.e., values greater than 0 remain positive, and values less than 0 remain negative). Consequently, as shown in Table 1 (linked), the final decoding accuracy remains unaffected, so we opt not to introduce an additional function.
>
> For BCELoss, the sigmoid function is necessary because BCELoss is designed for probabilistic outputs. The sigmoid ensures that the outputs lie within the [0,1] range, representing valid probability distributions. In contrast, MSELoss does not require such a transformation, as it directly optimizes the squared error.
>
> **Q2:** Thank you for your question. The term "Combined noise" was originally introduced by HiDDeN, and its purpose is to address the scenario where watermark images may encounter different types of distortions in real-world applications. Since it is difficult to predict which specific distortion may occur, the watermarking model needs to be trained to handle multiple types of noise simultaneously. In practice, during training, we employ a Combined Noise technique, where the model is exposed to a random noise layer in each mini-batch. This enables the model to learn robustness to multiple types of distortions at the same time. Additionally, during evaluation, we assess the model's robustness under different noise layers to demonstrate its general robustness. This approach is also widely used in watermarking models such as MBRS, CIN, and FIN.
>
> **Q3:** Thank you for your insightful suggestion. Following your advice, we experimented with directly using a few-step diffusion model as a noise layer for training. The results are presented in Table 2 (linked).
>
> As the number of steps (t) in the noise layer diffusion model increases, the robustness of the watermarking model against diffusion-based attacks also improves. However, with smaller t, the model’s performance remains inferior to surrogate method. We believe this is due to the difficulty of simulating large t attacks with smaller t. Additionally, using t=0.03 already demands substantial GPU memory and time, which limits our ability to explore larger values of t with our current resources.
>
> Despite these challenges, we believe your approach holds significant potential. Developing efficient few-shot diffusion models to simulate larger t attacks is an important direction for future research. However, this goes beyond the scope of our current paper. Our main contribution lies in providing a lightweight watermarking model that demonstrates state-of-the-art performance across various distortions. We are confident that our model can soon be integrated with future diffusion noise layers, and we believe it can as an efficient and effective framework for the watermarking community.
>
> **Q4:** Thank you for your valuable suggestion. We have included the results in Table 3 (linked). For your convenience, we summarize the findings below. Our method consistently performs best in most cases for the PRGAI attack, and it consistently ranks in the top two for the DiffPure attack.
>
> **Q5:**  Thank you for your insightful suggestion. We have included the watermarked image without an attack in Figure 1 (linked) and will update Figure 5 in the final version accordingly.
>
> **Q6:**  Thank you for your insightful suggestion. We conducted extensive experiments to evaluate the model's performance under stronger perturbations, and the results are provided in Table 4 (linked). For your convenience, we summarize the key findings below: Under stronger perturbations, both PH and DO methods remain competitive. Notably, DO outperforms other models across all distortions, except for Median Blur. This confirms the effectiveness of our methods and demonstrates that the robustness acquired from training with less perturbations generalizes well to more perturbations.

---

> > ### Comment · Reviewer_6BgQ · 2025-04-05
> >
> > After reviewing the authors' response, I suggest the paper's contribution valuable to the development of image watermarking overall. PH and DO provide comparable performance to prior works while using a lighter model design, which has potential to help future work improving performance.
> >
> > There are still several question I am considering:
> > - Regarding the PH approach versus $\text{tanh}$: Would the model perform better or worse if **trained** without PH but with a $\text{tanh}$ function clipping the output to the range $[-1, 1]$, with MSELoss directly applied to the output of the $\text{tanh}$ layer, compared to the PH approach?
> > - The paper mentions **Combined Noise** during testing time, which could be misinterpreted as cascading several noisy layers as a noisy channel. Readers would benefit from a clearer specification of the evaluation configuration.
> >
> > For future work, I suggest that:
> > 1. Testing DO with complex watermarking models such as FIN would strengthen the research, since DO is applicable to other watermarking models trained with MSELoss or BCELoss. Note that some distortions (style-transfer based, such as crayon, heavy color, etc.) are not evaluated in this paper; demonstrating performance improvement with FIN and its resilience to heavier distortions would provide valuable insights.
> > 2. Improving the robustness against diffusion-based watermark removal attacks is needed. Currently, Gaussian Noise + Median Filter works better than directly training with such attacks, as shown in the experimental results provided by the authors. However, it could be possible to address the challenge of incorporating a diffusion model with large edit strength ($t$) through a carefully designed training recipe (appropriate diffusion model, method to avoid multi-step denoising). Including discussion on adversarial attacks is crucial for robust watermarking.

---

> > > ### Author Response · Authors · 2025-04-09
> > >
> > > **We have included some results in the anonymous link:**
> > >
> > > https://anonymous.4open.science/r/Rebuttal-ICML-8393/Round2_Rebuttal_ICML2025_%208393.pdf
> > >
> > > **Question 1:**
> > > As shown in Table 1(linked), applying a `tanh` function to constrain the decoder output indeed improves performance compared to using MSELoss alone. However, the performance remains inferior to our PH approach.
> > >
> > > Our interpretation is as follows: while restricting the decoder’s output domain (e.g., via `tanh`) does help mitigate the impact of extreme values, it does **not** alleviate the influence of $L_{\text{inflation}}$
> > >  on already correctly decoded bits—similar to the use of `sigmoid` under BCE loss (see Appendix A.2). Without the PH block, the backbone alone must directly bear the burden of $L_{\text{inflation}}$, which consumes model capacity without contributing to decoding accuracy. In contrast, PH effectively offloads this burden, leading to better overall performance.
> > >
> > > **Question 2:**
> > > Thank you for your advice. In the final version, we will revise Section 4 (Experiments)  to provide a clearer and more detailed description of the noise layers used during testing.
> > >
> > > **Suggestion 1:**
> > > Thank you for the valuable suggestion. We provide additional results in Table 2(linked) to evaluate the generalization of DO to complex watermarking models such as FIN.
> > >
> > > We first applied DO to the original FIN architecture (8 INN blocks with 32-channel layers) and trained it using the same Combined Noise setting as our main paper. During testing, we evaluated its robustness against 6 white-box distortions and 4 black-box distortions mentioned in the FIN paper.
> > >
> > > As shown in Table 2, using DO or MSE on the original FIN leads to comparable results across all distortions. We attribute this to the fact that the original FIN already over-parameterizes for MSELoss, and its INN block design is near-optimal. Thus, even when DO relieves parameter pressure, performance gain is limited.
> > >
> > > To further test DO’s strength, we reduced the model size while preserving the INN block structure. Specifically, we built a lightweight "small FIN" by using 2 INN blocks and reducing the intermediate channel size from 32 to 2. We then trained small FIN using both MSE and DO losses.
> > >
> > > As shown in the table, small FIN trained with MSE suffers significant degradation under both white-box and black-box distortions. In contrast, DO remarkably preserves performance, achieving results close to the original FIN. Notably, under the Sketch distortion, DO outperforms MSE by more than **3%**, demonstrating DO's effectiveness in enabling robust performance under reduced capacity and challenging distortions.
> > >
> > > **Suggestion 2:**
> > > Thank you for the valuable and forward-looking suggestions.
> > >
> > > Diffusion-based Attacks:
> > >
> > > Our contribution to defending against diffusion-based watermark removal lies in the proposal of using **Gaussian Noise + Median Filter**, which has proven to be an **efficient and effective** strategy. While we agree that a carefully designed training recipe for diffusion models (e.g., appropriate model choice and avoiding multi-step denoising) is promising, this direction goes beyond the core scope of this work.
> > >
> > > Our main focus is to bridge the gap between commonly used decoding losses and actual decoding goals. We propose two generally applicable methods (DO and PH) that enable **lightweight models** to achieve **top-tier robustness and accuracy** across a wide range of distortions. The technical design of **lightweight AIGC-editing simulation layers** for robust training under diffusion-based attacks is an emerging area. To our knowledge, the most relevant recent work is the CVPR 2025 paper **OmniGuard: Hybrid Manipulation Localization via Augmented Versatile Deep Image Watermarking**.
> > >
> > > Adversarial Attacks:
> > >
> > > Regarding adversarial attacks, we explored adversarial training using **PGD attacks**. During training, we adopted PGD with parameters: `eps = 8/255`, `alpha = 2/255`, `iters = 5`, `norm = linf`. For evaluation, we used a stronger PGD with `iters = 1000`. As shown in Table 3(linked), while adversarial training slightly reduces robustness under the 6 white-box distortions, it **significantly enhances robustness** against adversarial attacks.
> > >
> > > This reveals a **robustness trade-off** between traditional distortions and adversarial attacks. In practice, users can balance this by tuning the weight between adversarial loss and decoding loss during training. Designing more comprehensive adversarial training strategies to mitigate this trade-off will be a promising future direction. To achieve provable robustness, one could explore random smoothing methods (**e.g. Certified adversarial robustness via randomized smoothing**).
> > >
> > > ---
> > > Thank you for the multiple rounds of comments. They have been helpful in improving the quality of our work. We have made every effort to add the necessary experiments and provide clear explanations, and we hope our responses address your concerns.

---

### Decision · Program_Chairs · 2025-05-01

**Decision:**

Accept (poster)

**Comment:**

This paper introduces a novel and practical framework for deep learning-based watermarking that aims to improve decoding efficiency and robustness while significantly reducing model complexity. The main innovations—Decoding-Oriented Surrogate Loss (DO) and Detachable Projection Head (PH)—are both conceptually sound and well-motivated, addressing a known gap in current loss formulations for watermark decoding. The manuscript offers strong empirical support through comprehensive evaluations across various distortions and benchmarking against state-of-the-art models. Despite some concerns from the reviewers, such as some type of distortions not evaluated in the experiments and some important hyperparameters not sufficiently clear, the contributions are meaningful and timely for the watermarking community.